# A general esterolysis strategy for upcycling waste polyesters into high-value esters

Minghao Zhang[1,2], Yunkai Yu[1,2], Buxing Han [ID][3] & Qingqing Mei [ID][1,2,4] ✉

The upcycling of waste polyesters into high-value chemicals offers a sustainable and economically viable solution to the global plastic waste crisis. Herein, we report a general esterolysis strategy for the efficient depolymerization of polyesters to produce high-value ester products, utilizing a broad range of esters, including carboxylates, carbonates, and C/Si/Ti/P-based esters. Using the 1-ethyl-3-methylimidazolium acetate as a highly effective catalyst, polyethylene terephthalate is selectively converted into dimethyl terephthalate and ethylene carbonate with remarkable yields of 99% and 90%, respectively. Mechanistic studies reveal that methanol, generated in situ via the 1-ethyl-3-methylimidazolium acetate-catalyzed hydrolysis of dimethyl carbonate, drives the cleavage of C−O ester bonds in polyethylene terephthalate. This strategy demonstrates broad applicability, achieving high conversion efficiencies across various mixed and colored commercial waste polyesters. The energy efficiency and versatility of this approach establish a transformative route to diverse high-value esters, advancing the development of circular plastic economies and sustainable chemistry.

Plastics, due to their exceptional material properties and convenience in manufacturing, have become integral to various industrial processes globally[1,2]. However, the extensive use of non-degradable, fossil fuel-derived plastics has led to severe environmental pollution and heightened concerns about the depletion of finite natural resources[3–6]. Polyethylene terephthalate (PET) is one of the most commercially prevalent plastics, with ~95% of its production transformed into waste within just 1 year of use[6–11]. Current recycling efforts primarily involve thermomechanical processes, which result in lower-grade polymers with diminished properties, thus limiting the applicability of this recycling method[12–14]. Thus, there is an urgent need for more efficient methods to upgrade and recycle waste PET into high-value products. Chemical recycling, mainly including solvolysis, hydrogenolysis, and pyrolysis, has emerged as a promising approach for converting waste plastics into valuable products[15–24]. In particular, solvolysis and co-solvent-assisted solvolysis, enables the breakdown of PET into its monomeric constituents: ethylene glycol (EG) and terephthalic acid

(TPA) or its derivatives such as bis(2-hydroxyethyl) terephthalate (BHET) and dimethyl terephthalate (DMT), which can then be used to manufacture new high-quality plastics[16,25–27] (Fig. 1A). These approaches have been extensively studied for their high potential in closed-loop recycling[24–26].

Despite its promise, solvolysis remains constrained by several persistent challenges, foremost among them being thermodynamic limitations that often lead to incomplete depolymerization[28]. To shift the equilibrium toward product formation, solvolysis typically relies on the use of large excesses of nucleophilic solvents, particularly alcoholysis. For instance, methanolysis commonly requires 20 to 50 equivalents of methanol to achieve satisfactory conversion (Table S1). Although co-solvent-assisted strategies can enhance depolymerization efficiency, they do not substantially reduce the overall demand for nucleophilic reagents[29,30]. The use of such excessive active nucleophilic reagents not only increases energy consumption but also promotes undesirable side reactions. Alcohols, for instance, are

[1]State Key Laboratory of Soil Pollution Control and Safety, Zhejiang University, Hangzhou, China. [2]Institute of Environment Science and Technology, College of Environmental and Resource Sciences, Zhejiang University, Hangzhou, Zhejiang, China. [3]Beijing National Laboratory for Molecular Sciences, CAS Laboratory of Colloid and Interface and Thermodynamics, Center for Carbon Neutral Chemistry, Institute of Chemistry, Chinese Academy of Sciences, Beijing, China. [4]Innovation Center of Yangtze River Delta, Zhejiang University, Jiaxing, Zhejiang, China. ✉e-mail: meiqq@zju.edu.cn

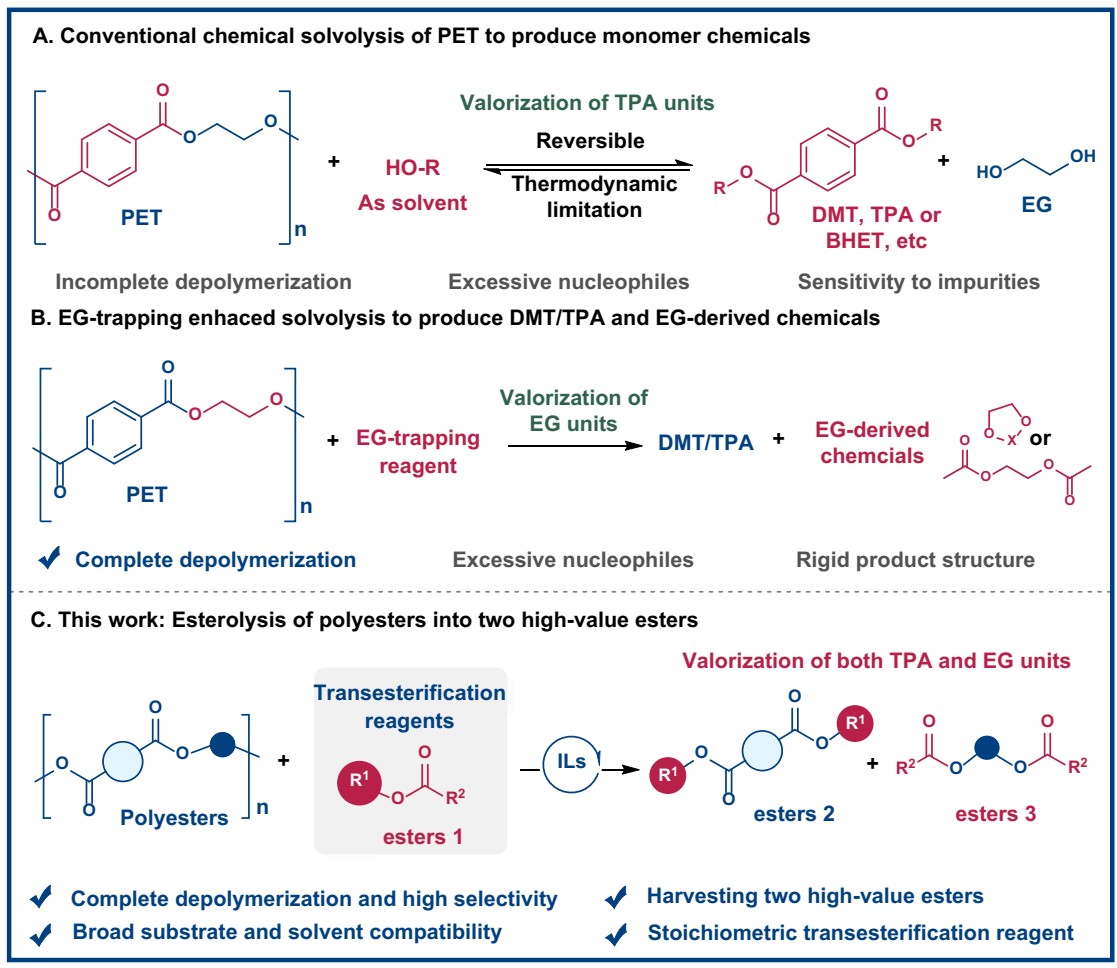

**Fig. 1 | Comparison of polyester solvolysis and esterolysis strategies.**
**A** Conventional chemical solvolysis of PET to produce monomer chemicals. **B** EG-trapping enhanced solvolysis to produce DMT/TPA and EG-derived chemicals.

**C** This work: Esterolysis of polyesters into two high-value esters. TPA terephthalic acid, DMT dimethyl terephthalate, BHET bis(2-hydroxyethyl) terephthalate, EG ethylene glycol, ILs ionic liquids.

prone to condensation under harsh reaction conditions, as exemplified by the formation of diethylene glycol from ethylene glycol during glycolysis[31]. Moreover, certain alcohols, such as allyl alcohol, are highly toxic and flammable, raising significant safety and environmental concerns.

To address these issues, various EG-trapping strategies have been developed to convert EG into value-added derivatives such as ethylene glycol diacetate[32,33], arylboronic esters[34], and ethylene carbonate[28,35] (Fig. 1B). However, these trapping agents often possess rigid molecular structures, limiting the diversity of accessible products, and still require substantial amounts of alcohol. Furthermore, both conventional solvolysis and these emerging approaches primarily focus on valorizing a single PET-derived monomer, either EG or TPA, with limited success in simultaneously upgrading both components. Developing integrated strategies that enable the concurrent, value-added transformation of both monomeric units is essential for fully leveraging the chemical potential of PET and improving the economic and environmental viability of plastic recycling.

In response to these challenges, we propose the concept of esterolysis for waste polyester conversion (Fig. 1C). Esterolysis is an exchange reaction in which esters act both as alkylating agents and transesterification partners, enabling the transformation of polyesters into structurally diverse, value-added esters. Unlike traditional alcoholysis, where alcohols directly serve as nucleophiles that attack the carbonyl groups of PET, esterolysis involves esters that undergo hydrolytic decomposition to release alcohol in situ. These alcohols then participate in transesterification with PET-derived EG, while the ester moiety simultaneously couples with TPA units. This dual recombination yields two distinct high-value esters from a single depolymerization event. The strategy offers several key advantages: (1) it exhibits broad substrate compatibility, accommodating a wide range of polyesters and carboxylic acid derivatives, including carboxylates, carbonates, and stoichiometric C-, Si-, Ti-, and P-based esters, thereby enabling full valorization of PET building blocks; (2) it enhances the reaction thermodynamics, as the simultaneous formation of two distinct esters shifts the equilibrium toward product formation and promotes complete depolymerization, while also reducing the overall reagent requirement; and (3) it demonstrates excellent water tolerance. Unlike base-catalyzed alcoholysis, where water often induces undesired side reactions, esterolysis proceeds effectively in the presence of ambient moisture and can even benefit from small amounts of added water, which facilitates ester hydrolysis. This work introduces an efficient, scalable, and environmentally friendly esterolysis platform for converting waste PET into structurally diverse esters. By overcoming critical limitations of conventional solvolysis, such as equilibrium constraints, reagent excess, and water sensitivity, this method offers a promising route for sustainable chemical recycling and high-value utilization of plastic waste.

## Results

Esters, such as alkyl carboxylate and carbonate, serve as stable and efficient sources of alkyl groups, and have been extensively utilized to

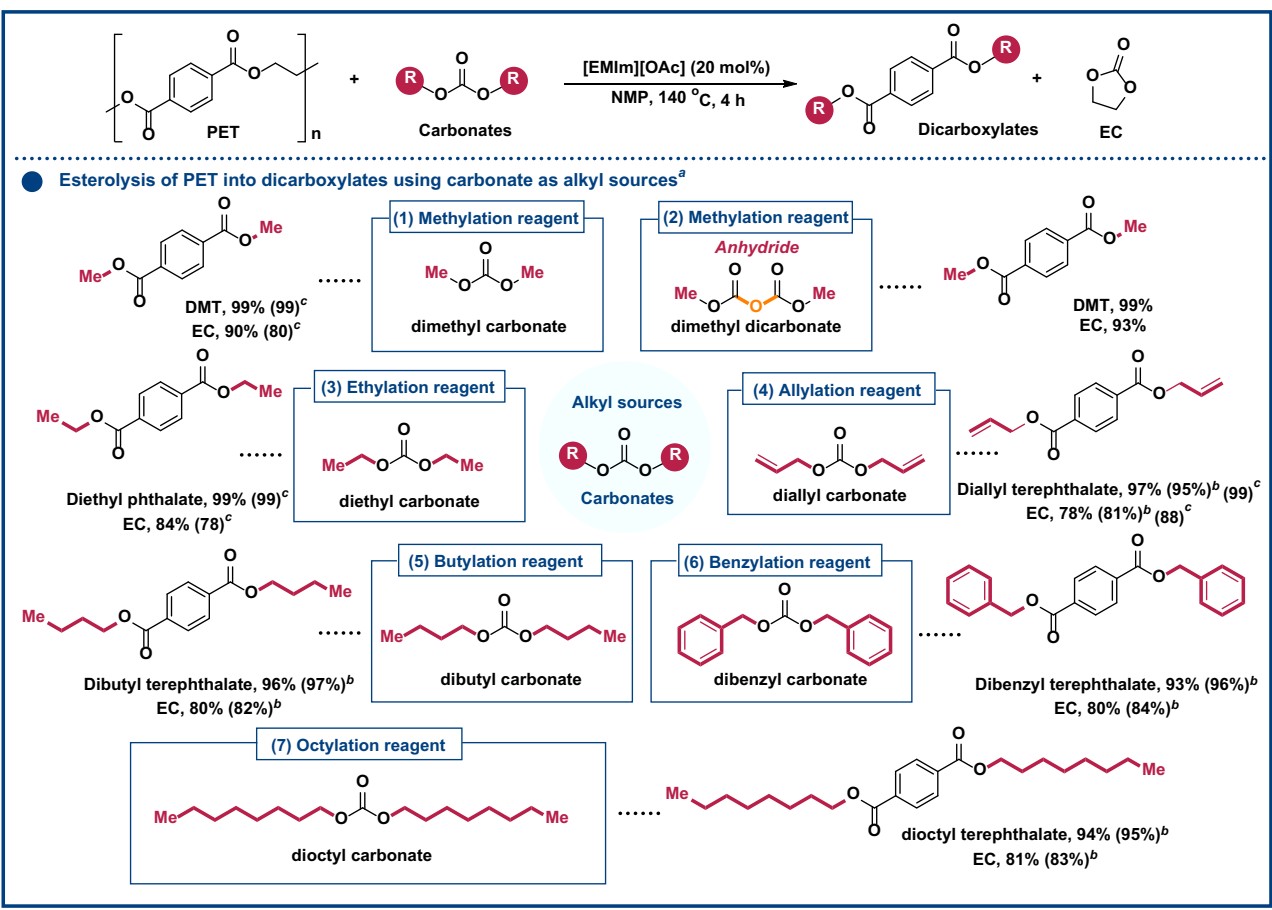

**Fig. 2 | Scope of carbonates.** [a]Standard reaction conditions: PET (1 mmol), carbonates (4 mmol), [EMIm][OAc] (20 mol%), and NMP (3 mL) at 140 °C for 4 h. [1]H NMR yields (600 MHz, CDCl$_3$, 298 K). [b]Carbonates (6 mmol). [c]Carbonates as both solvents and alkylating agents without NMP and external alcohols.

access alkylated chemicals[36–39]. For the modular conversion of PET into various alkylated terephthalates, different structured esters were selected as viable candidates for alkyl sources in the esterolysis of PET. We initiated our studies by investigating the reaction of PET with DMC (methyl source) without additional alcohol source. After extensive screening of the reaction parameters, DMT and EC were Obtained in 99% and 90% yield (GC and [1]H NMR yields, see Figs. S1 and S2 and Tables S2–S5 for details), respectively, under the optimal reaction conditions: PET (1 mmol), DMC (4 equiv), [EMIm][OAc] (20 mol%), NMP (3 mL) at 140 °C for 4 h (Fig. 2, entry 1 and Table S5). Notably, the strategy exhibits broad solvent compatibility, affording DMT in high yields (89-96%) across a range of common solvents instead of NMP, including DMSO, DMF, MeCN, 1,4-dioxane, acetone, and toluene. DMT and EC are both high-value ester compounds widely utilized across various industries[40,41]. For example, EC exhibits excellent solvency and is commonly used as a solvent in lithium-ion battery electrolytes, while DMT has been developed as a precursor feedstock in the production of advanced polyesters[40–42]. The [1]H NMR spectrum analysis of the products was depicted in Fig. S3. Additionally, dimethyl decarbonate possesses more nucleophilic carbonyl units for EG valorization, which enhanced the conversion of PET to DMT and EC, achieving yields of 99% and 93%, respectively (Fig. 2, entry 2). To obtain diverse alkylated terephthalates, we further examined carbonates with different functional group substitutions (ethyl, butyl, allyl, benzyl, and octyl) (Fig. 2, Nos. 3–7). Alkylated carbonates efficiently react with PET, delivering the corresponding functionalized terephthalates (diethyl phthalate, diallyl terephthalate, dibutyl terephthalate, and dibenzyl terephthalate) and EC, with yields of 93–99% and 78–93%, respectively. Notably, such carbonates, including DMC, diethyl carbonate, and diallyl carbonate, can readily function as both solvents and alkylating sources, enabling high yields of the corresponding terephthalate esters while simultaneously avoiding the use of toxic external alcohols and solvent like NMP.

This protocol was further applicable to a broad scope of carboxylates to produce DMT in moderate to excellent yields. Initially, dimethylmalonate (DMM) was employed as a methyl source in the esterolysis of PET, leading to the formation of DMT and EG-derived esters (Fig. 3 and Section 11 in Supplementary Information). These derivatives were produced from the reaction of malonic acid monomethyl ester with EG. By substituting different functional groups at the methylene position of carboxylates, these dicarboxylates 2b–2n enabled the esterolysis of PET into DMT with yields of 66–97%. The results showed that the electronic and steric effects of DMM derivatives had minimal impact on the yield of DMT, indicating good functional group tolerance. The adjustment of the carbon chain or structure of dicarboxylates 2o–2r was subsequently implemented, yielding DMT in 80–86% yield, catalyzed by [EMIm]Br. Other common alkyl or aromatic methylcarboxylates 2s–2y can also serve as sources of methyl groups, resulting in DMT yields ranging from 27% to 88%.

To further verify the applicability of the strategy, we subsequently explored the use of Si/Ti/C/P-based carboxylates as alkyl sources (Fig. 3C, D). These carboxylates were able to react with PET to produce a range of alkylated terephthalates, with yields varying from 54% to 94%. Notably, several intriguing noncyclic heteroatom-based and cyclic EG derivatives (including acetal, silicon, titanium, and phosphorus rings) were observed by LC-MS (See Section 11 in Supplementary Information for details). The effectiveness of this esterolysis strategy for upgrading waste PET was confirmed by the broad

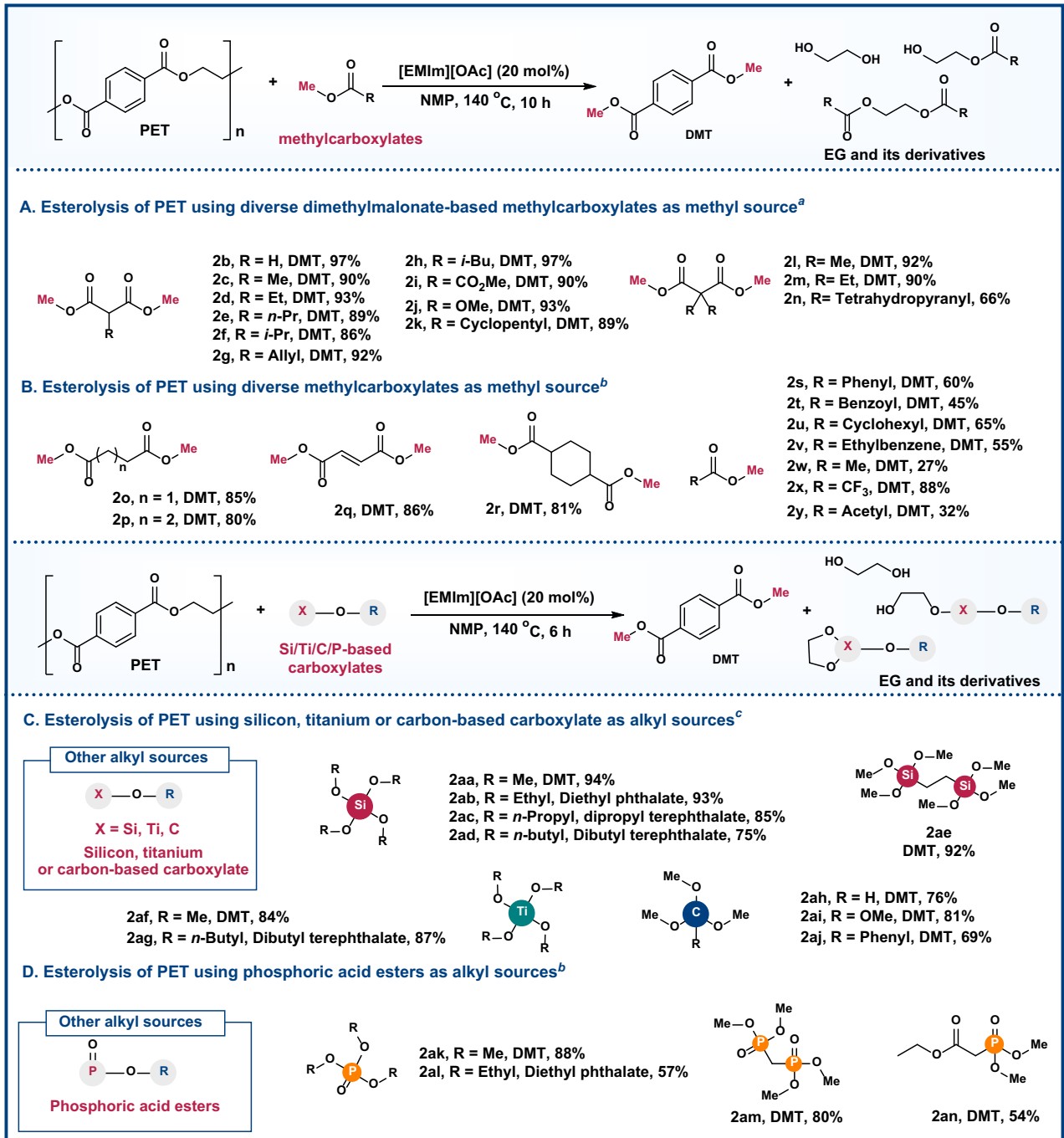

**Fig. 3 | Scope of different carboxylates. A** Esterolysis of PET using diverse dimethylmalonate-based methylcarboxylates as methyl source. **B** Esterolysis of PET using diverse methylcarboxylates as methyl source. **C** Esterolysis of PET using silicon, titanium, or carbon-based carboxylate as alkyl sources. **D** Esterolysis of PET using phosphoric acid esters as alkyl sources. [a]Standard reaction conditions: PET (1 mmol), esters (8 mmol), [EMIm][OAc] (20 mol%), and NMP (3 mL) at 140 °C for 10 h. [1]H NMR yields (600 MHz, CDCl₃, 298 K). [b]Standard reaction conditions: PET (1 mmol), carboxylates (15 mmol), [EMIm]Br (2 equiv), and NMP (3 mL) at 200 °C for 8 h. [c]Standard reaction conditions: PET (1 mmol), Si/Ti/C-based carboxylate (6 mmol), and NMP (3 mL) at 140 °C for 10 h.

substrate scope outlined, providing a promising and manageable value-added approach for PET upcycling.

Subsequently, we shifted our focus to exploring the waste plastic scope. As demonstrated in Fig. 4, a range of PET-based waste materials was examined. PET-based waste, including commercial woven tape, transparent film, woven mesh, used bottles, and non-woven fabric, were effectively degraded and converted to DMT and EC with yields of 94–98% and 84–91%, respectively (Fig. 4A). Gram-scale reactions were further conducted using 30 g and 50 g of raw PET bottle flakes in either

NMP or DMC as the reaction medium. After 4 h at 140 °C, the reactions yielded DMT in 97% and 98%, and EC in 81% and 85%, respectively. The produced DMT can be easily separated, and their purity was confirmed by [1]H NMR after recrystallization in MeOH, as depicted in Figs. S4–S6. We then analyzed the degradation of various commercially available polyester (PES, PEA, PEF and PLA) used across a range of industries (Fig. 4B). The DMC-involved esterolysis enabled conversion of these polyesters transformed into the corresponding esters (dimethyl succinate, dimethyl adipate, 2,5-furandicarboxylic acid and methyl

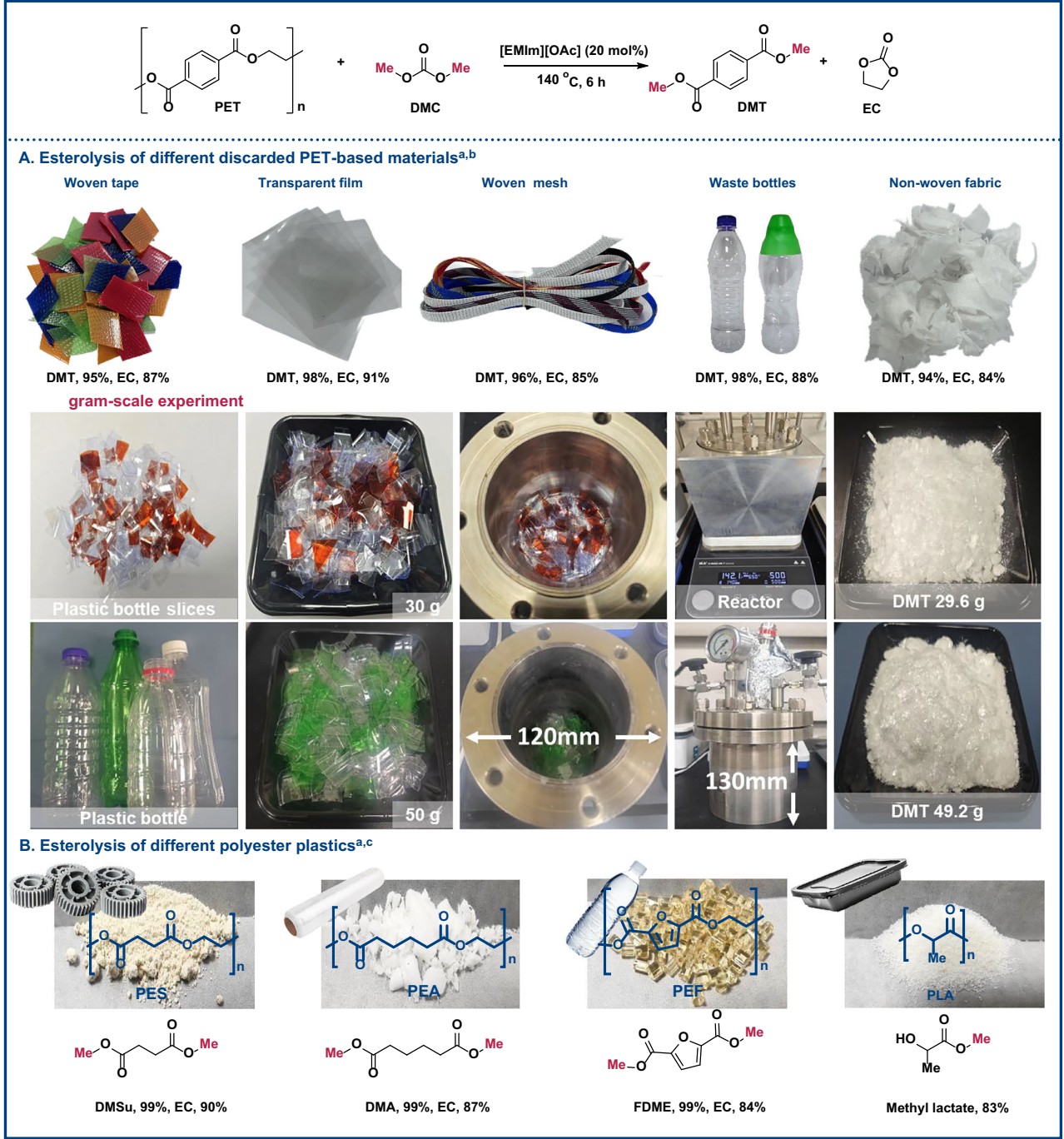

**Fig. 4 | Scope of waste PET materials and other polyesters. A** Esterolysis of different discarded PET-based materials. **B** Esterolysis of different polyester plastics. *ª*Standard reaction conditions: polyesters (1 mmol), DMC (4 mmol), [EMIm][OAc] (20 mol%), and NMP (3 mL) at 140 °C for 6 h. *ᵇ*GC yields. *ᶜ*¹H NMR yields (600 MHz, CDCl₃, 298 K).

lactate) with yields of 83–99% and 84–90% for EC, respectively. To address the issue of catalyst recyclability, we developed a polymeric ionic liquid catalyst (PIL-OAc), whose catalytic performance and reuse capability are summarized in Tables S6 and S7. These results demonstrate the effectiveness of this esterolysis strategy in achieving plastic waste valorization.

To elucidate the overall reaction mechanism and the factors influencing PET decomposition, we conducted a detailed kinetic investigation of PET esterolysis, as shown in Fig. S7. The results indicated that methanolysis alone was insufficient for efficient PET degradation, with depolymerization identified as the rate-limiting step.

To explore the initiation of this depolymerization, trace amounts of MeOH and H₂O were introduced into the system (Figs. S8 and S9). The results revealed that even small additions of MeOH markedly accelerated the esterolysis process, irrespective of whether DMC or DMM was used as the transesterification reagent. Similarly, low concentrations of H₂O enhanced the reaction efficiency in DMC-mediated systems and exhibited an even more pronounced effect in DMM-involved esterolysis. These findings indicate that both H₂O and in situ-generated MeOH are pivotal in initiating the depolymerization process. To validate this hypothesis, we examined the origin of MeOH in DMC-mediated PET esterolysis (Fig. 5). GC-MS analysis confirmed that

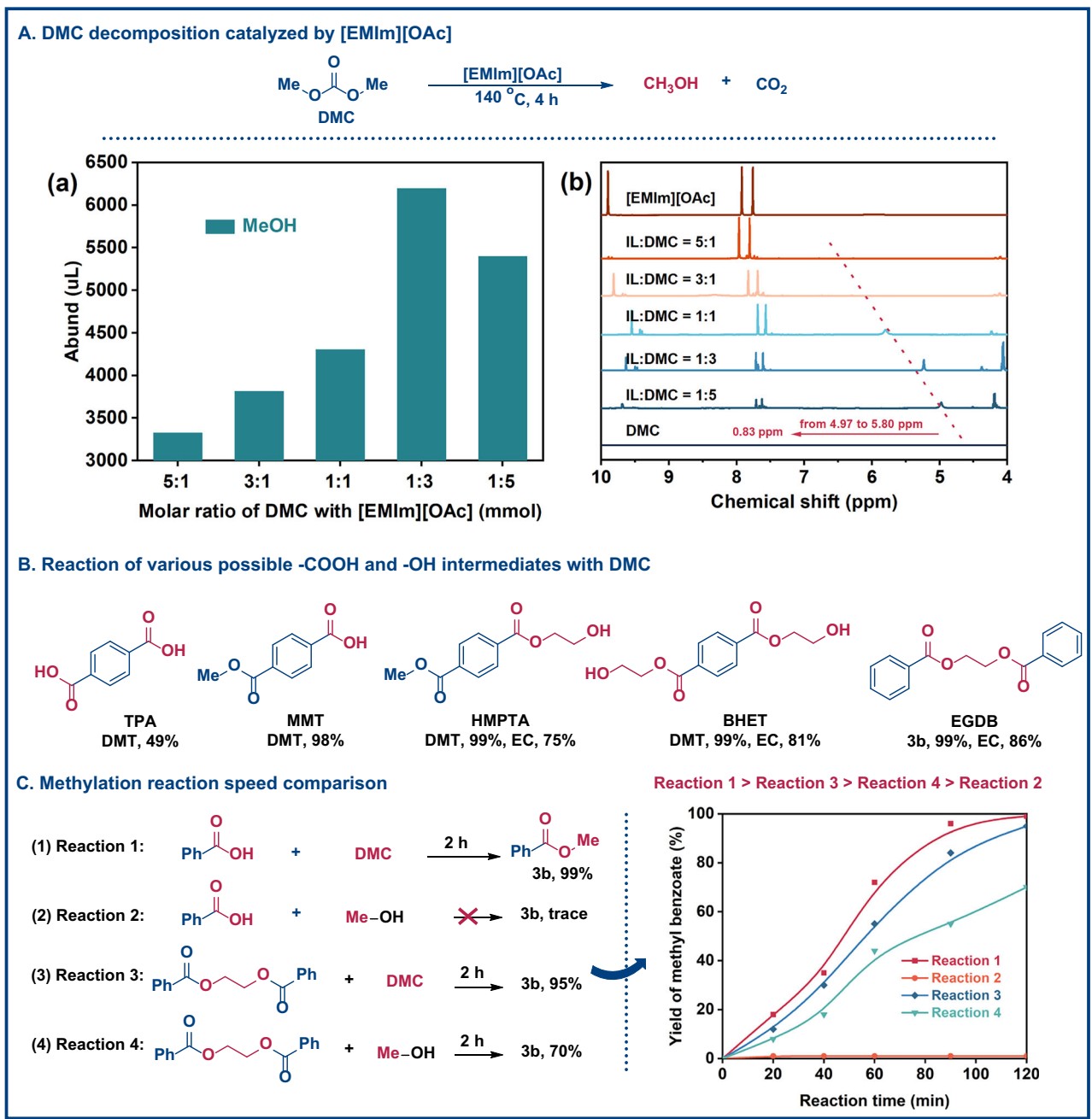

**Fig. 5 | Mechanistic study of PET depolymerization via esterolysis. A** DMC decomposition catalyzed by [EMIm][OAc]. (a) GC-MS analysis of MeOH production. (b) ¹H NMR analysis (600 MHz, DMSO-*d6*, 298 K) of MeOH production. **B** Reaction of various possible -COOH and -OH intermediates with DMC. Standard reaction conditions: possible monomer (1 mmol), DMC (4 mmol), [EMIm][OAc] (20 mol%), and NMP (3 mL) at 140 °C for 4 h. ¹H NMR yield (600 MHz, CDCl₃, 298 K). **C** Methylation reaction speed comparison. Standard reaction conditions: possible monomer (1 mmol), methylation reagents (4 mmol), [EMIm][OAc] (20 mol%), and NMP (3 mL) at 140 °C.

the hydrolysis of DMC, catalyzed by [EMIm][OAc], rather than the solvent NMP, was responsible for MeOH production (Figs. 5A and S10). A series of controlled experiments on the causes of PET decomposition emphasized the critical role of methanol production from DMC hydrolysis (Figs. S11 and S12). Notably, the concentration of MeOH increased in correlation with the amount of [EMIm][OAc] present in the reaction mixture. ¹H NMR spectroscopy further demonstrated significant hydrogen bonding between the hydroxyl group of MeOH and the acetate anion ([OAc]⁻) (Fig. 5B). As the ratio of [EMIm][OAc] to DMC increases, more stronger hydrogen bonds were gradually formed, leading to a downfield shift that eventually disappeared as the concentration of [EMIm][OAc] rises[43,44]. Specifically, the chemical shift

of the hydroxyl hydrogen of MeOH shifts downfield from 4.97 to 5.80 ppm when the molar ratio of [EMIm][OAc] to DMC changed from 1:5 to 1:1. The decomposition of esters to produce MeOH was also detected in the reaction mixtures of [EMIm][OAc] and DMM by ¹H and ¹³C NMR analysis (Fig. S13). Previous studies have demonstrated that the hydrogen bonds formed between ionic liquids and reactants facilitate the cleavage of C−O bond of esters in PET[44-47]. These interactions are illustrated in Fig. S14, where both the anions and cations of [EMIm][OAc] engage with the reactants (PET) and the intermediates (EG and MeOH). The hydrogen bonding between alcohols (EG, MeOH) and [EMIm][OAc] was further supported by FT-IR analysis (Fig. S15)[43,44]. These findings suggested that, under the catalysis of [EMIm][OAc],

trace amounts of $H_2O$ triggered the hydrolysis of DMC, resulting in the production of MeOH[48,49]. The produced MeOH then efficiently attacked the ester bond of PET, leading to the cleavage of the C–O ester bond (Figs. 5A, S12 and S13)[29,45,47,50].

To probe the intermediates following the initial depolymerization, we subsequently subjected plausible model reaction intermediates, including carboxylic acids, alcohols, and esters, to methylation reactions. The results demonstrated that these model monomers undergo efficient transesterification with DMC, yielding the corresponding methyl esters with 49–99% efficiency (Fig. 5B). Notably, carboxylic acids (TPA and MMT), derived from PET hydrolysis, reacted with DMC to afford DMT in yields of 49% and 99%, respectively, indicating that PET hydrolysis to carboxylic acid intermediates followed by consecutive methylation provides a possible and viable transformation pathway. Considering the importance of developing effective carboxylic acid methylation strategies, we explored the scope of methylation of various carboxylic acids with DMC (Fig. S16)[37]. This methylation strategy demonstrated excellent tolerance to both electronic and steric effects, with yields ranging from 90% to 99%.

To elucidate the potential transesterification reaction processes, we investigated the reaction kinetics of benzoic acid and ethane-1,2-diyl dibenzoate with methylating agents (MeOH or DMC). As illustrated in Fig. 5C, the reaction rate follows the order: reaction 1 > reaction 3 > reaction 4 > reaction 2 over the same period. Direct transesterification of benzoic acid with DMC exhibited the fastest conversion speed. The results demonstrated that, under the catalysis of [EMIm][OAc], DMC effectively facilitated the direct methylation of benzoic acid to yield 3b; conversely, MeOH does not react with benzoic acid (Fig. 5C, entries 1 and 2)[37]. Furthermore, the transesterification reaction between ethane-1,2-diyl dibenzoate and MeOH is less efficient than that using DMC (entries 3 and 4). This result suggested that DMC significantly accelerated the transesterification reaction by capturing the EG units to produce EC[28,35]. This result indicated that the produced MeOH primarily enhances the initial C–O band cleavage of ester groups in PET, while the subsequent PET conversion is predominantly driven by the transesterification reaction between DMC and EG. Based on our latest experimental data, we present a DMC-involved PET esterolysis mechanism facilitated by [EMIm][OAc], as described in Fig. S17. While methanolysis appears to be the predominant pathway, supporting evidence also points to a concurrent esterolysis route involving carboxylic acid esters, as illustrated in Fig. S18.

## Discussion

In this study, we presented an efficient esterolysis strategy for upcycling waste polyester into valuable esters via transesterification, offering an alternative approach for polyester valorization. By reacting PET with carboxylates, carbonates, and C/Si/Ti/P-based esters as alkyl sources, we produced two types of esters with high yields: terephthalates (93–99%) and ethylene carbonate (78–93%). Mechanistic studies revealed that methanol, generated via [EMIm][OAc]-catalyzed hydrolysis of methylcarboxylates, triggers C–O bond cleavage in PET esters, distinguishing this approach from conventional alcoholysis mechanisms. However, as a proof of concept, this study proposes a general ionic liquid catalyst without further optimization for specific reactions or catalyst recovery. While this work demonstrates a proof of concept using a general ionic liquid catalyst, further optimization for specific reactions and catalyst recyclability remains to be addressed. Future research should prioritize the design of highly efficient, task-specific, and easily recyclable heterogeneous catalysts to expand the applicability and enhance the sustainability of this approach. Overall, this facile strategy offers a conceptual framework for transforming waste polyesters into high-value esters, contributing to the molecular-level understanding and pathway design of polyester upcycling, and

supporting the broader vision of circular chemistry in addressing plastic waste challenges.

## Methods

### General procedure esterolysis of PET with DMC

PET powder (1 mmol), [EMIm][OAc] (20 mol%), DMC (4 mmol), and NMP (3 mL) were combined in a polytetrafluoroethylene rotor (25 mL), which was then hermetically sealed within a stainless-steel reactor. The reactor was stirred at 140 °C for 4 h with a rotational speed of 500 rpm on the heating plate. Following the completion of the reaction, the system was quenched by transferring the reactor into an ice bath. The liquid products were analyzed by GC and $^1H$ NMR (600 MHz, $CDCl_3$, 298 K) using mesitylene (1 mmol) as an internal standard (See Supplementary Information for details). These conditions were also applicable to the depolymerization of commercial PET waste and other polyesters

### General procedure esterolysis of PET with DMM

PET powder (1 mmol), [EMIm][OAc] (20 mol%), DMM (8 mmol), and NMP (3 mL) were combined in a polytetrafluoroethylene rotor (25 mL), which was then hermetically sealed within a stainless-steel reactor. The reactor was stirred at 140 °C for 10 h with a rotational speed of 500 rpm on the heating plate. Following the completion of the reaction, the system was quenched by transferring the reactor into an ice bath. The liquid products were analyzed by $^1H$ NMR (600 MHz, $CDCl_3$, 298 K) using mesitylene (1 mmol) as an internal standard. Other Si/Ti/C-based carboxylates were involved in PET depolymerization using the same method.

### General procedure esterolysis of PET with other methylcarboxylates

PET powder (1 mmol), [EMIm]Br (2 equiv), methylcarboxylates (15 mmol), and NMP (3 mL) were combined in a polytetra-fluoroethylene rotor (25 mL), which was then hermetically sealed within a stainless-steel reactor. The reactor was stirred at 200 °C for 8 h with a rotational speed of 500 rpm on the heating plate. Following the completion of the reaction, the system was quenched by transferring the reactor into an ice bath. The liquid products were analyzed by $^1H$ NMR (600 MHz, $CDCl_3$, 298 K) using mesitylene (1 mmol) as an internal standard.

### Characterization methods

The product yield was analyzed by gas chromatography (GC): Agilent 8860, with Agilent J&W HP-5 polysiloxane gas chromatography column; gas chromatography-mass spectrometry (GC-MS): Agilent 7890A/5975C GC/MSD, equipped with Agilent J&W HP-5 polysiloxane gas chromatography column. The $^1H$ NMR and $^{13}C$ NMR spectra were recorded at room temperature using a Bruker Avance-600 instrument ($^1H$ NMR frequency of 600 MHz, $^{13}C$ NMR frequency of 151 MHz). The NMR spectra of all products were in ppm, with reference to the solvent signal [$^1H$ NMR: CD(H)Cl$_3$ (7.26 ppm), $^{13}C$ NMR: CD(H)Cl$_3$ (77.00 ppm)]. The signal mode is expressed as: s, single; d, double; dd, double double; t, triple; m, multiple. Gas chromatography-mass spectrometry (GC-MS): Agilent 7890A/5975C GC/MSD with Agilent J&W HP-5 Polysiloxane GC Column. Liquid Chromatograph Mass Spectrometer (LC-MS): Waters ACQUITY Premier/Xevo G2-XS was used in chromatographic experiments. The mobile phases were water (A) and acetonitrile (B). The linear gradient pro-grams were as follows, 0 min 5%B; 4 min 95%B; 7 min 95%B; 10 min 5%B; Sample injection volume, 1 μL; Column oven temperature, 40 °C; Flow rate, 0.4 mL min$^{-1}$. Scanning Electron Microscopy (SEM) was conducted using a Hitachi SU8010 microscope at an acceleration voltage of 3 kV. Images were captured in secondary electron mode at 500× magnifications under high vacuum conditions.

## Data availability

Data relating to experimental procedures, mechanistic studies, and all other data supporting the findings are available within the article and its Supplementary Information. All data are available from the corresponding author upon request.

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

## Acknowledgements

This work was supported by the National Natural Science Foundation of China (grant numbers 22376183 and 22209146 to Q.M.) and Key Research and Development Program of Zhejiang Province (grant numbers 2024C03112 to Q.M.).

## Author contributions

Q.M. conceived the research. M.Z. conducted most of the experiments, characterization, and data analysis. M.Z. wrote the manuscript primarily. Y.Y. conducted part of the characterization experiments. All authors (M.Z., Y.Y., B.H., and Q.M.) contributed to discussions and manuscript review.

## Competing interests

The authors declare no competing interests.
