## [Transparent Peer Review file · Nature Communications]

A General Esterolysis Strategy for Upcycling Waste Polyesters into High-Value Esters

Corresponding Author: Professor Qingqing Mei

Version 0:

Reviewer comments:

Reviewer #1

(Remarks to the Author)

Review of the Manuscript: " A General Esterolysis Strategy for Upcycling Waste Polyesters into High-Value Esters "

After evaluating this manuscript, I consider that it presents a relevant contribution to the chemical recycling of polyesters. The work is well-structured, with a clear justification of the problem, a solid methodology, and convincing results. However, I have some comments and questions that could help strengthen the research:

Main Comments:

1) Regarding to the mechanism explanation, in the section on in situ methanol production, it is mentioned that methanol is key to cleaving the C–O bond in polyesters. Have possible side effects of methanol accumulation in the reaction been considered?

2) In the experiment table, it is observed that long-chain carbonates require 6 mmol instead of 4 mmol. Could the authors explain why this occurs? Is it due to lower reactivity or the need to balance the reaction?

3) In Figure S3, the presence of free ethylene glycol is mentioned, but not integrated. Could this indicate that the conversion is incomplete or that there are unconsidered byproducts? Additionally, the integration of this product should also be considered for yield determinations.

4) The authors have studied 13 derivatives of dimethyl malonate, but I missed a deeper discussion on the influence of these compounds (as methyl group sources) on the reaction yield. Could the enol/ketone equilibrium of these dimethyl malonates have any impact on the efficiency of the esterolysis?

Overall, the manuscript is solid and presents an innovative approach. I appreciate the opportunity to review it and hope that my comments are useful in strengthening the work.

Reviewer #2

(Remarks to the Author)

The authors represent an interesting approach to addressing plastic waste through an esterolysis strategy, and the reported reactivity is noteworthy. The mechanistic insights and broad applicability to various polyesters further contribute to the existing body of knowledge in this field. The key idea is that esters and carbonates (in particular) can be used instead of traditional alcohols, leading to two carbonyl-based products, sometimes both of value. The process is demonstrated on an impressive array of substrates including post consumer waste.

The strategy is novel, but borrows heavily from existing work detailing fast depolymerization of PET in solvents - including

NMP (Ind. Eng. Chem. Res. 2018, 57, 48, 16239) and the use of dimethyl carbonate to trap EG (Green Chem. 2021, 23, 9412, ACS Mater. Au 2024, 4, 3, 335). The latter reference is cited as the very last reference in the paper, instead of the pair being acknowledged as a key advance upon which this work is built. Several other key advances are ignored in the introduction. A more comprehensive literature review would provide stronger context for the work.

While the work represents a nice idea and is carefully done, there are several problems associated with the study. This field is inherently a practical one - the substrates involved are not new or complex, so to be of broad significance, a study in this area needs to be potentially applicable on scale the scale of tonnes, be inexpensive, sustainable and economical. Else, it will never find application. Without being potentially applicable, all that remains is a nice way to do some very simple chemistry that would not excite the broader readership of the journal.

Specifically

- The catalyst loading of 20% ionic liquid is significantly high, making the process economically (and perhaps environmentally) impractical.
- The use of NMP solvent at 3 mL/mmol is excessive, further impacting the sustainability and scalability of the approach. Remember this chemistry will have to be carried out on tonne scale and generate reasonably inexpensive products.
- Excess of the 'alkylating agents' are used (3-8 equiv usually). These are usually expensive relative to the alcohols that would be used in the industrially applicable alcoholysis chemistry this study is hoping to complement. In addition, these agents are invariably synthesised from the alcohols themselves, leaving one to wonder what the real advantage here is over straightforward, faster and cheaper alcoholysis.

Despite the high catalyst loading and large solvent volume, the depolymerisation of PET remains far too slow considering the 140C reaction time, which raises further concerns about the efficiency of the method.

The mechanistic work is convincingly done.

In summary, this is a nice idea which does add to the field, but it forgets the inherent need for plastic recycling methodologies to be simple, fast, economical and sustainable to be actually useful on scale. Without that applicability, we would recommend that this study is not suitable for publication in this journal, however it is definitely publishable elsewhere.

Reviewer #3

(Remarks to the Author)

The author has introduced a novel ester hydrolysis pathway based on an ester exchange mechanism and validated its potential applications. While the innovation is promising, several issues need to be addressed:

1. The manuscript utilizes NMP as the solvent, but separating it from the product during recovery is challenging. Additionally, the catalyst used in the reaction and the product, ethylene carbonate (EC), are not effectively separated or recovered, making it impractical for waste plastic recycling. The author must address this complication.
2. Can the catalyst be recycled in this reaction? This is crucial for waste plastic recycling.
3. Why was [Bmim]Br chosen over [Bmim]OAc in the experiments detailed in Figure 3b? Figure 3 explores a broad range of reactants, yet the yields of the corresponding ester exchange products are not indicated. Please include this information.
4. As mentioned by the author, the OAc anion initiates the activation of the ester hydrolysis process by activating methanol produced from DMC decomposition. However, this resembles typical alcoholysis mechanisms. The author should further elaborate on the distinctions between these pathways to illustrate that the proposed ester hydrolysis pathway is genuinely innovative.

Minor errors: capitalization issues in reference titles. Please review and rectify these errors.

Version 1:

Reviewer comments:

Reviewer #1

(Remarks to the Author)

Having revised the updated version of the manuscript, all the points I initially indicated have been satisfactorily addressed. I therefore consider that the manuscript could be accepted.

Reviewer #2

(Remarks to the Author)

Considerable effort has been made to address concerns, however while the effort is commendable, the main point here seems to be lost.

These are not interesting molecules (the products) which can be easily synthesized in the lab. The fact that esterolysis is 'new' does not make it interesting to the broad readership of Nature Communications unless it looks like this might be a practical methodology at scale. Then it is very interesting. This does not mean that all the scale-up issues need to be worked out now - however there are so many barriers to adoption in any practical sense here (given how cheap PET and DMT are) that the excess reagents, high loadings of catalyst etc make the process difficult to see as being useful.

For example, the statement in the rebuttal 'in contrast, esterolysis in our study operates efficiently with 3-8 equivalents of ester, thus requiring a lower overall reagent input' is fine in the lab, but are we going to use 3-8 equivalents of ester (made themselves from alcohols and acids) on multi-tonne scale?

Using DMC is nice, but also well known in biodiesel production, and its more expensive than methanol. The reactions were also less efficient in it. The solvent can be recycled, but you can clearly see MeOH in the recycled product by ¹H NMR (not exactly a sensitive technique).

The catalyst loading problem is not solved in my view. The authors have provided a technicality which actually takes the methodology further from being useful. They make a polymer bound catalyst, requiring investment, reagents etc. This is fine. But ca. 25 wt% of this is required to catalyze the reaction, at higher temperatures than before. They recycle it 3 times, and EC yields are already dropping on the first recycle and both DMT and EC are lower on the third.

This is clearly not a strategy that would be adopted at scale. What industry would invest in making a catalyst (more complex than simple systems available for DMT synthesis) which is already less efficient after one recycle? This is a technicality - it allows the authors to claim that a recyclable catalyst is available, but which in fact does not make the system any more practical. Heating at high temperatures for long times is still a problem too.

While the authors have made attempts to remedy the issues, I feel the process as it stands is just not practical enough (by a wide margin) to be of general interest.

Reviewer #3

(Remarks to the Author)

We have carefully reviewed the revised manuscript and the authors' responses to our comments. Overall, the authors have addressed the majority of the concerns raised in the previous review, and the revisions have improved the clarity and rigor of the work.

Regarding Question 3 (on quantification in substrate scope evaluation), while the authors were unable to resolve the issue of product quantification definitively, they have provided reasonable justification for their approach and clarified their perspective. Although further experimental validation would have strengthened the study, their explanation is acceptable in the current context.

Given the authors' efforts in addressing the reviewers' feedback and the manuscript's scientific merit, we recommend acceptance of the paper in its present form.

Reviewer #1: After evaluating this manuscript, I consider that it presents a relevant contribution to the chemical recycling of polyesters. The work is well-structured, with clear justification of the problem, a solid methodology, and convincing results. However, I have some comments and questions that could help strengthen the research:

Response: We appreciate the reviewer for the positive comments and suggestions for improving this manuscript.

Main Comments:

Comments 1: Regarding to the mechanism explanation, in the section on in situ methanol production, it is mentioned that methanol is key to cleaving the C–O bond in polyesters. Have possible side effects of methanol accumulation in the reaction been considered?

Response: We thank the reviewer for the thoughtful comment regarding the potential impact of methanol accumulation on the reaction mechanism. To address this, we systematically investigated how varying amounts of methanol affect the esterolysis process.

As shown in **Figure R1**, we introduced different amounts of methanol (0.1–1 mmol) into the reaction and monitored the outcome after an initial 40-minute period. The results revealed that the addition of 0.6 mmol methanol led to the fastest depolymerization, with complete PET conversion within 40 minutes. Under these conditions, high yields of DMT (99%) and EC (85%) were achieved. These findings suggest that moderate amounts of methanol enhance the reaction at the early stage.

Figure R1. Effect of methanol dosage on the reaction after the initial 40 minutes.

To further examine the effects of methanol accumulation over prolonged reaction times, we performed control experiments with larger methanol additions (0.05–2 mL, ca. 1–50 mmol) after 4 hours (**Figure R2**). When methanol was increased to 0.4 mL (~10 mmol), full PET conversion was still maintained with consistently high DMT yields (99%). However, further increasing methanol to 2.0 mL led to a decline in both DMT yield and overall depolymerization efficiency. Notably, EC yield decreased steadily with rising methanol concentration. This decline can be attributed to the formation of excessive hydrogen bonding between methanol and [EMIm][OAc], which likely disrupts the critical interactions between [EMIm][OAc], DMC, and EG. This interference may inhibit the forward progress of transesterification and methanolysis reactions, ultimately resulting in diminished depolymerization performance.

Figure R2. Effect of methanol dosage on the reaction after 4 hours.

In conclusion, while moderate methanol levels enhance depolymerization by promoting methanolysis pathways, excessive accumulation hinders the reaction by disrupting critical intermolecular interactions. Under our optimized conditions, however, methanol formation remains at a favorable level, and no adverse effects on reaction efficiency were observed. Thus, methanol accumulation does not adversely affect the reaction under the established operating conditions.

Comments 2: In the experiment table, it is observed that long-chain carbonates require 6 mmol instead of 4 mmol. Could the authors explain why this occurs? Is it due to lower reactivity or the need to balance the reaction?

Response: We thank the reviewer for pointing out this important observation. As shown in **Table R1**, increasing the amount of DMC facilitates the conversion of PET to DMT and EC. This enhancement is attributed to more methylating agents and nucleophilic attack sites for EG, thereby driving the reaction equilibrium toward EC formation. During the expansion of our substrate scope to include long-chain carbonates, we found that using 4 equivalents led to slightly lower EC yields compared to DMC. This difference is likely due to increased steric hindrance and/or reduced reactivity of the longer-chain alcohols formed in situ, which may be less efficient in driving the transesterification equilibrium. To compensate for this and improve EC yields, we increased the carbonate loading to 6 equivalents. In the original manuscript, we reported only the optimal condition (6 equivalents), but as noted by the reviewer, the reactions using 4 equivalents also gave acceptable yields of both terephthalate esters and EC. In response to this helpful suggestion, we have now included additional data and an explanation in the revised manuscript to provide a more complete picture.

Table R1. Comparison of methylation reagent dosage.

Entry	Catalyst	loading	Temperature	Solvent	DMC (mmol)	DMT Yield (%) ^b	EC Yield (%) ^b
1	[Emim][OAc]	20 mol%	140 °C	NMP	1	39	12
2	[Emim][OAc]	20 mol%	140 °C	NMP	2	57	33
3	[Emim][OAc]	20 mol%	140 °C	NMP	3	94	78
4	[Emim][OAc]	20 mol%	140 °C	NMP	4	99	90
5	[Emim][OAc]	20 mol%	140 °C	NMP	5	99	89

Catalytic system and reaction conditions explanation. ^a Standard reaction conditions: PET (0.192g, 1 mmol structural unit), DMC (X mmol), [Emim][OAc] (20 mol%), and NMP (3 mL) at 140°C for 4 h. ^b Yields were determined by GC analysis with mesitylene as internal standard.

Here are the details:

Revised manuscript, Page 3:

Figure R3. Scope of carbonates. ^a Standard reaction conditions: PET (1 mmol), carbonates (4 mmol), [EMIm][OAc] (20 mol%), and NMP (3 mL) at 140 °C for 4 h. ¹H NMR yields. ^b Carbonates (6 mmol). ^c Carbonates as both solvents and alkylating agents without NMP and external alcohols.

Comments 3: In Figure S3, the presence of free ethylene glycol is mentioned, but not integrated. Could this indicate that the conversion is incomplete or that there are unconsidered byproducts? Additionally, the integration of this product should also be considered for yield determinations.

Response: We appreciate the reviewer's insightful comment. Quantitative analyses by GC and NMR using mesitylene as an internal standard revealed a consistent DMT yield of 99%. The absence of any insoluble or suspended solid further indicated complete PET conversion. To determine whether the depolymerization process produced any unaccounted EG-derived byproducts such as dioxane, 2-hydroxyethyl methyl carbonate, or 1,2-dimethoxycarbonyloxyethane, a more comprehensive analysis was performed for the ^1H NMR spectra of the product mixture. As shown in the revised **Figure R4**, the spectrum confirmed that PET was fully converted into DMT and EC, with yields of 99% and 90%, respectively. During the reaction process, the produced EG effectively in-situ reacts with DMC via transesterification in the system to produce EC. The product mixture contained unreacted EG (~3-4%) and 2-hydroxyethyl methyl carbonate (~5%), as well as EC, with no detectable formation of the 1,2-dimethoxycarbonyloxyethane.

Figure R4. ^1H NMR spectrum (600 MHz, CDCl_3 , 298 K) of reaction mixture with mesitylene (1 mmol) as internal standard.

In response to the reviewer's suggestion, we have updated the supplementary information (Page 6) to incorporate the integration of free EG and its derivatives into the product distribution analysis, thereby ensuring a more accurate representation of the reaction outcome.

Here are the details:

Revised Supporting information, Page 14: This ^1H NMR spectrum, as described in Figure S3, indicated that PET was fully converted into DMT and EC with a yield of 99% and 90%, respectively. The product mixture contained unreacted EG (~3-4%) and 2-hydroxyethyl methyl carbonate (~5%), as well as EC, with no detectable formation of the by-product 1,2-dimethoxycarbonyloxyethane.

Comments 4: The authors have studied 13 derivatives of dimethyl malonate, but I missed a deeper discussion on the influence of these compounds (as methyl group sources) on the reaction yield. Could the enol/ketone equilibrium of these have any impact on the efficiency of the esterolysis?

Response: We thank the reviewer for this thoughtful comment. In exploring the scope of dimethyl malonate derivatives, we found that mono-substituted alkyl groups with varying steric hindrance (-Me, -Et, -*n*-Pr, -*i*-Pr, -*i*-Bu, and -cyclopentyl), as well as electron-donating or electron-withdrawing substituents (-allyl, -CO₂Me, and -OMe), had minimal impact on the yield of DMT, indicating good functional group tolerance. In these cases, high yields were obtained, ranging from 86% to 97%. Moreover, disubstituted alkyl groups such as -Me and -Et were also compatible with this system, affording DMT in 92% and 90% yields, respectively. A slight decrease in yield was observed with the tetrahydropyranyl-substituted derivative, likely due to steric rigidity and reduced conformational flexibility. Overall, these findings underscore the broad functional group tolerance of our strategy.

To address the reviewer's concern whether the enol/ketone equilibrium of dimethyl malonate derivatives might influence the esterolysis reaction, DFT calculations were performed to evaluate the conversion of dimethyl malonate into methyl 3-hydroxy-3-methoxyacrylate. The calculated Gibbs free energy change (ΔG) for this transformation is 14.0 kcal/mol. The relatively high ΔG indicates that the formation of the enol form is thermodynamically unfavorable under standard conditions. This suggests that the enol form contributes negligibly to the overall reaction pathway, and the esterolysis proceeds predominantly via the keto form.

Figure R5. DFT calculations the enol/ketone equilibrium of dimethyl malonate derivatives

In response to the reviewer's suggestion, we have included a summary in the revised manuscript describing the impact of different functional group substitutions on dimethyl malonate derivatives with respect to the yield of DMT.

Here are the details:

Revised manuscript, Top of the Page 3: The results showed that the electronic and steric effects of DMM derivatives had a minimal impact on the yield of DMT, indicating good functional group tolerance.

Reviewer #2: The strategy is novel but borrows heavily from existing work detailing fast depolymerization of PET in solvents - including NMP (Ind. Eng. Chem. Res. 2018, 57, 48, 16239) and the use of dimethyl carbonate to trap EG (Green Chem. 2021, 23, 9412, ACS Mater. Au 2024, 4, 3, 335). The latter reference is cited as the very last reference in the paper, instead of the pair being acknowledged as a key advance upon which this work is built. Several other key advances are ignored in the introduction. A more comprehensive literature review would provide stronger context for the work.

While the work represents a nice idea and is carefully done, there are several problems associated with the study. This field is inherently a practical one - the substrates involved are not new or complex, so to be of broad significance, a study in this area needs to be potentially applicable on scale the scale of tonnes, be inexpensive, sustainable and economical. Else, it will never find application. Without being potentially applicable, all that remains is a nice way to do some very simple chemistry that would not excite the broader readership of the journal.

Response: We sincerely thank the reviewer for the thoughtful and constructive feedback. We appreciate your recognition of the novelty and careful execution of our work, and we fully agree that a thorough contextualization is essential to emphasize its broader significance and practical relevance. Following your recommendation, we have expanded the Introduction to include several key advances in PET depolymerization involving solvent-mediated processes and trapping reagents. Specifically, we now more prominently acknowledge the foundational contributions of solvent systems such as NMP (Ind. Eng. Chem. Res. 2018, 57, 48, 16239) and the use of dimethyl carbonate for EG trapping (Green Chem. 2021, 23, 9412; ACS Mater. Au 2024, 4, 3, 335). These works are now cited and discussed as important precursors that helped shape our approach, rather than being placed peripherally in the reference list. We agree that the proper positioning of such prior art is critical for accurately framing the novelty of our work, and we thank the reviewer for pointing this out.

To further strengthen the practical relevance and address concerns regarding scalability, sustainability, and economic feasibility, we have made several substantive improvements in this revision:

1) Recyclable Catalyst: We developed and utilized a recyclable polymeric ionic liquid catalyst (PIL-OAc) in place of conventional [EMIm][OAc], addressing limitations in catalyst reuse and downstream

separation.

2) Green Reaction Media: We verified a series of low-toxicity, low-boiling-point carbonates as both solvents and alkylating agents to replace external toxic alcohols and high-boiling-point solvents

3) Scale-Up Demonstration: We carried out a 50 g scale-up reaction under the optimized conditions to demonstrate the feasibility of larger-scale implementation.

4) Solvent Recovery: We verified efficient solvent recovery and reuse, highlighting the potential for process circularity and industrial translation.

These developments directly respond to the reviewer's concerns and enhance the practical scope of our strategy. We believe they help bridge the gap between conceptual novelty and real-world applicability. We are once again grateful for your detailed and insightful comments, which have greatly improved the scientific depth and contextual framing of our manuscript. We hope that these comprehensive revisions address your concerns and clarify the broader value of our work to both academic and industrial audiences.

Here are the details:

Revised manuscript, Page 1: To shift the equilibrium toward product formation, solvolysis typically relies on the use of large excesses of nucleophilic solvents, particularly alcoholysis. For instance, methanolysis commonly requires 20 to 50 equivalents of methanol to achieve satisfactory conversion (Table S1). Although co-solvent-assisted strategies can enhance depolymerization efficiency, they do not substantially reduce the overall demand for nucleophilic reagents^{29,30}.

Middle of page 2: To address these issues, various EG-trapping strategies have been developed to convert EG into value-added derivatives such as ethylene glycol diacetate^{32,33}, arylboronic esters^{28,34}, and ethylene carbonate^{27,35}. However, these trapping agents often possess rigid molecular structures, limiting the diversity of accessible products, and still require substantial amounts of alcohol.

Bottom of Page 2: The strategy offers several key advantages: (1) it exhibits broad substrate compatibility, accommodating a wide range of polyesters and carboxylic acid derivatives, including carboxylates, carbonates, and stoichiometric C-, Si-, Ti-, and P-based esters, thereby enabling full valorization of PET building blocks; (2) it enhances the reaction thermodynamics, as the simultaneous formation of two new esters shifts the equilibrium toward product formation and promotes complete

depolymerization, while also reducing the overall reagent requirement; and (3) it demonstrates excellent water tolerance. Unlike base-catalyzed alcoholysis, where water often induces undesired side reactions, esterolysis proceeds effectively in the presence of ambient moisture and can even benefit from small amounts of added water, which facilitates ester hydrolysis.

Notably, the strategy exhibits broad solvent compatibility, affording DMT in high yields (89-96%) across a range of common solvents instead of NMP, including DMSO, DMF, MeCN, 1,4-dioxane, acetone, and toluene.

Top of Page 2: Notably, such carbonates, including DMC, diethyl carbonate, and diallyl carbonate, can readily function as both solvents and alkylating sources, enabling high yields of the corresponding terephthalate esters while simultaneously avoiding the use of toxic external alcohols and solvent like NMP.

Bottom of Page 4: To demonstrate the potential for industrial scalability, gram-scale reactions were conducted using 30 g and 50 g of raw PET bottle flakes in either NMP or DMC as the reaction medium. After 4 hours at 140 °C, the reactions yielded DMT in 97% and 98%, and EC in 81% and 85%, respectively.

Bottom of Page 4: To address the issue of catalyst recyclability, we developed a polymeric ionic liquid catalyst (PIL-OAc), whose catalytic performance and reuse capability are summarized in Tables S6 and S7. These results further enhance the practical potential of this esterolysis strategy for sustainable plastic waste valorization.

Revised supporting information, Page 16, Figure S5: To validate the effectiveness of this strategy, we conducted a scale-up experiment on a 50 g scale using PET bottle flakes, with DMC serving as both the solvent and methanol source, without the need for additional methanol. As shown in Figure S5a, the post-reaction mixture could be readily separated via solid–liquid separation, yielding a liquid product phase and crude DMT. The crude DMT was purified by recrystallization to afford high-purity DMT (98%), while the liquid phase was efficiently recycled by rotary evaporation to recover DMC, leaving only trace amounts of methanol (Figure S5b and 5c).

Page 18, Table S6: In response to the challenges associated with recycling [EMIm][OAc], we explored the use of poly(ionic liquid)s and successfully designed and synthesized a novel catalyst, PIL-OAc, using p-divinylbenzene, zinc acetate, and 1-vinyl-3-methylimidazolium acetate as monomers, with

AIBN as the initiator and MeCN as the solvent. To optimize the reaction conditions, a preliminary screening of catalyst loading, temperature, and solvent was performed. The results demonstrated that the synthesized PIL-OAc exhibited excellent catalytic activity, enabling complete depolymerization of PET. Notably, when NMP was used as the solvent at 160 °C, DMT and EC were obtained in yields of 99% and 78%, respectively. Notably, efficient PET conversion can be achieved using low-boiling-point solvents such as acetonitrile, yielding DMT and EC in 99% and 81%, respectively.

Page 19, Table S7: The PIL-OAc catalyst can be readily separated from the reaction system after the reaction. Even after three recycling cycles, it still enables efficient conversion of PET to DMT with a high yield of 97%. Although a slight decrease in DMT yield was observed upon extended recycling, the successful implementation of this process provides strong evidence supporting the recyclability of this catalytic system.

[27] Tanaka, S., Sato, J. & Nakajima, Y. Capturing ethylene glycol with dimethyl carbonate towards depolymerisation of polyethylene terephthalate at ambient temperature. *Green Chem.* 23, 9412-9416 (2021).

[28] Zhang, M. H. et al. Converting waste PET into dimethyl terephthalate and diverse boronic esters under metal-free conditions. *Green Chem.* 26, 11132-11139 (2024).

[29] Tang, J. et al. Mechanistic insights of cosolvent efficient enhancement of PET methanol alcohololysis. *Ind. Eng. Chem. Res.* 62, 4917-4927 (2023).

[30] Liu, B. et al. Ultrafast homogeneous glycolysis of waste polyethylene terephthalate via a dissolution-degradation strategy. *Ind. Eng. Chem. Res.* 57, 16239-16245 (2018).

[31] Lee, T., Peng, Y. K., Lee, H. L. & Pratama, D. E. Chemical recycling development of poly(ethylene terephthalate) by glycolysis and cooling crystallization with water. *Ind. Eng. Chem. Res.* 62, 19873-19883 (2023).

[32] Peng, Y. T. et al. Acetolysis of waste polyethylene terephthalate for upcycling and life-cycle assessment study. *Nat. Commun.* 14, 3249 (2023).

[33] Luo, Y. J., Sun, J. Y. & Li, Z. Rapid chemical recycling of waste polyester plastics catalyzed by recyclable catalyst. *Green Chem. Eng.* 5, 257-265 (2024).

[34] Zhang, M. et al. Full valorisation of waste PET into dimethyl terephthalate and cyclic arylboronic esters. *Appl. Catal. B Environ. Energy* 352, 124055 (2024).

[35] Tanaka, S. et al. Depolymerization of polyester fibers with dimethyl carbonate-aided methanolysis. *ACS Mater. Au* 4, 335-345 (2024).

Comments 1: The catalyst loading of 20% ionic liquid is significantly high, making the process economically (and perhaps environmentally) impractical.

Response: We thank the reviewer for the valuable comment regarding the high catalyst loading of [EMIm][OAc] and its implications for economic and environmental feasibility. As the reviewer rightly notes, the recovery and reuse of homogeneous ionic liquids pose significant challenges in practical applications, primarily due to issues such as incomplete separation, potential contamination of products, and degradation or loss of catalytic activity.

To address this concern, we developed and tested a recyclable polymeric ionic liquid catalyst (PIL-OAc) based on 1-vinyl-3-methylimidazolium acetate, zinc acetate, and *p*-divinylbenzene (*Polym. Degrad. Stabil.* 2022, 199, 109905). This polymeric design retains the catalytic activity of [EMIm][OAc] while enabling straightforward separation and reuse.

Under optimized conditions (160 °C, NMP solvent), PIL-OAc efficiently catalyzed the depolymerization of PET to deliver DMT and EC in high yields of 99% and 78%, respectively (**Table R2**). Moreover, the catalyst was easily recovered via phase separation and maintained its structural integrity and activity over multiple recycling cycles. Importantly, these signals do not indicate structural degradation of the catalyst. Recycling experiments demonstrated the catalyst's robustness: DMT yield remained above 97% after three cycles, and EC yield exhibited only a moderate decline (**Table R3**).

These findings demonstrate the practical feasibility and improved sustainability of the esterolysis process when implemented with a recyclable PIL-based catalyst. While further optimization is underway to enhance catalyst longevity and reduce production costs, this work represents a meaningful step toward scalable and economically viable plastic upcycling, exhibiting high application potential of our esterolysis strategy.

Table R2. PILs-catalyzed PET esterolysis condition with DMC^a

Entry	Cat.	Load.	Temp.	Sol.	DMT Yield (%) ^b	EC Yield (%) ^b
1	PIL-OAc	0.05g	150 °C	NMP	95	70
2	PIL-OAc	0.05g	160 °C	NMP	99	78
3	PIL-OAc	0.05g	170 °C	NMP	99	45
4	PIL-OAc	0.1g	160 °C	NMP	99	60
5	PIL-OAc	0.03g	160 °C	NMP	97	71
6	PIL-OAc	0.05g	160 °C	MeCN	99	81
7	PIL-OAc	0.05g	160 °C	Acetone	99	39
8	PIL-OAc	0.05g	160 °C	DMC	99	52

Catalytic system and reaction conditions exploration. ^a Standard reaction conditions: PET (0.192g, 1 mmol structural unit), DMC (4 mmol), and solvent (3 mL) at 140 °C for 4 h. ^b Yields were determined by ¹H NMR analysis with mesitylene as internal standard.

Table R3. Recyclability Test of PIL-OAc

Cycle Number	Load.	Temp.	Sol.	DMT Yield (%) ^b	EC Yield (%) ^b
1	0.05g	160 °C	NMP	99	72
2	0.05g	160 °C	NMP	97	63

Catalytic system and reaction conditions exploration. ^a Standard reaction conditions: PET (0.192g, 1 mmol structural unit), DMC (4 mmol), and solvent (3 mL) at 140 °C for 4 h. ^b Yields were determined by ¹H NMR analysis with mesitylene as internal standard.

Here are the details:

Revised manuscript, Bottom of Page 4: To address the issue of catalyst recyclability, we developed a polymeric ionic liquid catalyst (PIL-OAc), whose catalytic performance and reuse capability are summarized in Tables S6 and S7. These results further enhance the practical potential of this esterolysis strategy for sustainable plastic waste valorization.

Revised supporting information, Page 18, Table S6: In response to the challenges associated with recycling [EMIm][OAc], we explored the use of poly(ionic liquid)s and successfully designed and synthesized a novel catalyst, PIL-OAc, using *p*-divinylbenzene, zinc acetate, and 1-vinyl-3-methylimidazolium acetate as monomers, with AIBN as the initiator and MeCN as the solvent. To optimize the reaction conditions, a preliminary screening of catalyst loading, temperature, and solvent was performed. The results demonstrated that the synthesized PIL-OAc exhibited excellent catalytic activity, enabling complete depolymerization of PET. Notably, when NMP was used as the solvent at 160 °C, DMT and EC were obtained in yields of 99% and 78%, respectively. Notably, efficient PET conversion can be achieved using low-boiling-point solvents such as acetonitrile, yielding DMT and EC in 99% and 81%, respectively.

Page 19, Table S7: The PIL-OAc catalyst can be readily separated from the reaction system after the reaction. Even after three recycling cycles, it still enables efficient conversion of PET to DMT with a high yield of 97%. Although a slight decrease in DMT yield was observed upon extended recycling, the successful implementation of this process provides strong evidence supporting the recyclability of this catalytic system.

Comments 2: The use of NMP solvent at 3 mL/mmol is excessive, further impacting the sustainability and scalability of the approach. Remember this chemistry will have to be carried out on tonne scale and generate reasonably inexpensive products.

Response: We thank the reviewer for this valuable comment regarding solvent usage and its implications for large-scale implementation. We fully agree that excessive use of high-boiling-point solvents such as NMP can hinder both the sustainability and economic feasibility of the process.

While NMP was initially selected for mechanistic studies due to its high solvating power and compatibility with polyester depolymerization, it is by no means essential to the reaction. In fact, we evaluated a broad range of solvents and found that alternatives such as DMF and DMSO also promoted esterolysis effectively, yielding 89% and 96% DMT, respectively. More importantly, numerous low-boiling, readily recoverable solvents, including acetonitrile, benzotrifluoride, 1,4-dioxane, toluene, cyclohexane, THF, acetone, and DMC, were also found to be compatible, affording DMT in yields of 80–99% (**Table R4**). Particularly promising are carbonate solvents such as diethyl carbonate and diallyl carbonate, which can serve dually as solvents and transesterification agents, enabling the formation of high-value terephthalate esters without external solvents. These volatile solvents are easily recovered and reused, significantly reducing environmental burden and enhancing cost-efficiency.

To further validate the scalability and sustainability of the esterolysis strategy, we performed 50 g scale-up experiments using PET bottle flakes with DMC as both the solvent and reagent, eliminating the need for external methanol and high-boiling-point solvents such as NMP. As shown in **Figure R6**, the reaction mixture underwent straightforward solid-liquid separation to yield a liquid product phase and crude DMT. The latter was purified by recrystallization to afford high-purity DMT (98%), while the liquid phase was efficiently recycled via rotary evaporation to recover DMC. ^1H and ^{13}C NMR analysis of the recovered DMC confirmed its integrity, with only trace amounts of residual methanol detected, thus supporting the recyclability of the solvent in this system. Furthermore, our polymeric ionic liquid catalyst (PIL-OAc) is compatible with several of these recyclable solvents, including DMC and acetonitrile (**Table R2**), facilitating downstream separation and enhancing overall process robustness.

In summary, although NMP was initially employed for optimization, our strategy is not limited

to high-boiling solvents. The system performs effectively with inexpensive, low-boiling, and recyclable alternatives, and the successful scale-up experiment highlights the method's practical scalability and sustainability.

Table R4. Comparison of reaction solvent

Entry	Catal.	Load.	Temp.	Sol.	DMT Yield (%) ^b
1	[Emim][OAc]	20 mol%	140 °C	DMSO	89
2	[Emim][OAc]	20 mol%	140 °C	NMP	99
4	[Emim][OAc]	20 mol%	140 °C	DMF	96
5	[Emim][OAc]	20 mol%	140 °C	Benzotrifluoride	94
6	[Emim][OAc]	20 mol%	140 °C	1,4-Dioxane	92
7	[Emim][OAc]	20 mol%	140 °C	MeCN	91
8	[Emim][OAc]	20 mol%	140 °C	Toluene	91
10	[Emim][OAc]	20 mol%	140 °C	Cyclohexane	85
11	[Emim][OAc]	20 mol%	140 °C	THF	80
12	[Emim][OAc]	20 mol%	140 °C	Acetone	92
13	[Emim][OAc]	20 mol%	140 °C	Chlorobenzene	9
14	[Emim][OAc]	20 mol%	140 °C	Bromobenzene	-
15	[Emim][OAc]	20 mol%	140 °C	Chloroform	-
16	[Emim][OAc]	20 mol%	140 °C	DMC	99

Catalytic system and reaction conditions exploration. ^a Standard reaction conditions: PET (1 mmol), DMC (4 mmol), [Emim][OAc] (20 mol%), and solvent (3 mL) at 140 °C for 4 h. ^b Yields were mesitylene by GC analysis with mesitylene as internal standard.

Figure R6. Solvent recovery experiments: (a) Separation of solvent and recovery of products; (b) ^1H NMR spectrum of the recycled DMC; (c) ^{13}C NMR spectrum of the recycled DMC.

Here are the details:

Revised manuscript, bottom of Page 2: Notably, the strategy exhibits broad solvent compatibility, affording DMT in high yields (89-96%) across a range of common solvents instead of NMP, including DMSO, DMF, MeCN, 1,4-dioxane, acetone, and toluene.

Top of Page 3: Notably, such carbonates, including DMC, diethyl carbonate, and diallyl carbonate, can readily function as both solvents and alkylating sources, enabling high yields of the corresponding terephthalate esters while simultaneously avoiding the use of toxic external alcohols and solvent like NMP.

Bottom of Page 4: To demonstrate the potential for industrial scalability, gram-scale reactions were conducted using 30 g and 50 g of raw PET bottle flakes in either NMP or DMC as the reaction medium. After 4 hours at 140 °C, the reactions yielded DMT in 97% and 98%, and EC in 81% and 85%, respectively.

Revised supporting information, Page 16, Figure S5: To validate the effectiveness of this strategy, we conducted a scale-up experiment on a 50 g scale using PET bottle flakes, with DMC serving as

both the solvent and methanol source, without the need for additional methanol. As shown in Figure S5a, the post-reaction mixture could be readily separated via solid–liquid separation, yielding a liquid product phase and crude DMT. The crude DMT was purified by recrystallization to afford high-purity DMT (98%), while the liquid phase was efficiently recycled by rotary evaporation to recover DMC, leaving only trace amounts of methanol (Figure S5b and 5c).

Comments 3: Excess of the 'alkylating agents' are used (3-8 equiv usually). These are usually expensive relative to the alcohols that would be used in the industrially applicable alcoholysis chemistry this study is hoping to complement. In addition, these agents are invariably synthesised from the alcohols themselves, leaving one to wonder what the real advantage here is over straightforward, faster and cheaper alcoholysis.

Response: We sincerely thank the reviewer for this insightful comment. We would like to clarify that the aim of this work is not to replace conventional alcoholysis, but to introduce *esterolysis* as a conceptually distinct and complementary strategy for polyester upcycling. While alcoholysis is typically optimized for monomer recovery, esterolysis enables direct conversion of waste polyesters into value-added functional esters using ester reagents as alkyl donors. To address the reviewer's concern regarding reagent excess and cost, we note that alcoholysis processes in literature often use larger excesses of alcohol (commonly 20-50 equivalents) to drive equilibrium-limited depolymerization (see **Table R5**). In contrast, esterolysis in our study operates efficiently with 3-8 equivalents of ester, thus requiring a lower overall reagent input.

Table R5. Methanolysis of PET with reported catalysts.

Entry	Catalysts	MeOH:PET	Co-solvent	Temp. (°C)	T (h)	Conv. (%)	Yield (%)	Ref.
1	CO ₂	n(MeOH:PET) = 36:1	-	220	0.83	100	79.1	[1]
2	AlP	n(MeOH:PET) = 76:1	Toluene	200	2	-	88.5	[2]
3	Pb(AC) ₂ + Zn(AC) ₂	n(MeOH:PET) = 16.8:1	-	120	2	97.8	97.8	[3]
4	ZnO nanodispersions	n(MeOH:PET) = 36:1	-	170	0.25	97	95	[4]
5	K ₂ CO ₃	n(MeOH:PET) = 50:1	DCM	25	24	100	93.1	[5]
6	Orange peel ash	n(MeOH:PET) = 47.4:1	-	200	1	-	79	[6]
7	OPA@Fe ₂ O ₃	n(MeOH:PET) = 49:1	-	200	1	100	83	[7]
8	LiOMe	n(MeOH:PET) = 23.4:1	DMC	65	5	-	91	[8]
9	Bamboo leaf ash	n(MeOH:PET) = 49.4:1	-	200	2	100	78	[9]
10	Ti _{0.5} Si _{0.5} O ₂	n(MeOH:PET) = 46.8:1	-	160	2	100	98.2	[10]
11	Calcined sodium silicate	n(MeOH:PET) = 30:1	-	200	0.5	100	95	[11]
12	MgO/NaY	n(MeOH:PET) = 36:1	-	200	0.5	99	91	[12]
13	ChCl/Zn(OAc) ₂	n(MeOH:PET) = 15:1	MeCN	170	1	100	90.1	[13]
14	[HDBU][Im]	n(MeOH:PET) = 4.8:1	-	140	3	100	75	[14]
15	PIL-Zn ²⁺	n(MeOH:PET) = 24:1	-	170	1	100	90.3	[15]
16	[HO ₃ S-(CH ₂) ₃ -NEt ₃]Cl[ZnCl ₂] _{0.67}	n(MeOH:PET) = 24:1	-	195	0.5	-	78.4	[16]
17	DBN/Phenol	n(MeOH:PET) = 18:1	-	130	1	100	95.3	[17]
18	[BMIm][OAc]	n(MeOH:PET) = 24:1	-	150	4	88.5	41.7	[18]
19	[EMIm][OAc]	n(DMC:PET) = 4:1	NMP	140	4	100	99	This work

While there is some overlap in product outcomes between esterolysis and traditional alcoholysis, and we acknowledge that certain esters used in our study may currently be more expensive than commodity alcohols, esterolysis presents distinct advantages in specific chemical contexts. Taking PET as an example, the use of esterolysis is strategically motivated by broader benefits in safety, thermodynamic efficiency, reaction control, and sustainability. These advantages justify its exploration as a versatile and potentially scalable alternative. We outline five key benefits below:

(1) Functional Scope Beyond Monomer Recovery: Unlike alcoholysis targeting monomer recovery, esterolysis enables selective conversion of *both* terephthalic acid (TPA) and ethylene glycol (EG) units into high-value esters. For example, by selecting functional esters (e.g., Si-, Ti-, C-, or P-based), EG can be directly transformed into organometallic esters without the need for additional

capturing agents. Many of these transformations cannot be achieved via direct alcoholysis because the corresponding alcohols are unstable or synthetically inaccessible. Moreover, metal alkoxides are generally less reactive with alcohol under mild conditions but can engage in transesterification with esters via ligand exchange, allowing for greater selectivity and efficiency.

- (2) Thermodynamic Control and Equilibrium Disruption:** Ester reagents in our system serve a dual role, as alkylating agents for TPA and as effective traps for EG, suppressing its side reactions and shifting the equilibrium toward depolymerization. This enables efficient PET conversion with significantly lower ester loadings than conventional alcoholysis, which typically requires 20-50 equivalents of alcohol (**Table R5**) to overcome equilibrium limitations. These challenges are further amplified when using bulky or unreactive alcohols. In contrast, our esterolysis strategy bypasses such constraints, producing a broad range of terephthalate esters without external alcohols. Additionally, the in situ-generated EG undergoes further esterification to yield valuable co-products, establishing a cascade process that improves both reaction efficiency and product value.
- (3) Improved Selectivity via Controlled Reactivity:** Esters are widely used as chemically stable "protected forms" of reactive alcohols in organic synthesis due to their relatively lower reactivity. This property can be leveraged to suppress undesired side reactions during depolymerization. Under controlled conditions, esters can gradually release alcohol, allowing for better control over reaction pathways and improved selectivity. In conventional PET alcoholysis, the in situ-generated EG is prone to condensation, forming undesired diethylene glycol byproducts that lower product yield, compromise selectivity, and complicate downstream separation. In contrast, in our esterolysis system, both the EG released from PET and the alcohols derived from ester decomposition are rapidly consumed in subsequent reactions, preventing their accumulation. This cascade behavior minimizes side reactions and enhances the overall efficiency, selectivity, and operational stability of the process.
- (4) Green process and safer feedstocks:** Many ester compounds, such as carbonates (e.g., DMC, diethyl carbonate, propylene carbonate, diallyl carbonate) and carboxylate esters (e.g., methyl benzoate, methyl formate, methyl acetate, and methyl propionate), can serve dually as both low-toxicity alkyl source and green solvents, reducing the need for additional volatile organic compounds (VOCs). In contrast, the corresponding alcohols are often more toxic, corrosive to equipment, and difficult to handle safely on an industrial scale. For instance, diallyl carbonate

exhibits significantly lower toxicity than allyl alcohol and has been widely used in polyester and polymer synthesis. In our work, diallyl carbonate serves as both the allylating agent and the solvent, enabling the efficient production of diallyl terephthalate with a yield of 99% (**Figure R7**) without external alcohols or NMP. This dual functionality not only simplifies the reaction system but also improves the atom economy and enhances the overall sustainability of the process.

Figure R7. Comparison of green and safety in PET depolymerization strategy

(5) Raw material applicability and cost-effectiveness. Transesterification can directly utilize macromolecular esters from natural or synthetic sources such as plant oils or polyesters without the need for prior hydrolysis into alcohol or carboxylic acids, thereby streamlining the overall process. Actually, a variety of naturally occurring esters, including triglycerides, isoamyl acetate, phenethyl acetate, and wax esters, are abundant in plants and fruits and are not necessarily derived from alcohol precursors. These abundant naturally occurring esters, along with inexpensive industrial ester-rich by-products such as fatty acid methyl ester residues generated during petroleum refining, represent promising alkoxy sources for transesterification. With the development of more efficient catalytic systems, natural or low-purity industrial waste esters could serve as cost-effective alkoxy donors for polyester transesterification, replacing high-purity alcohols and thereby reducing reaction costs while enhancing resource efficiency.

We fully acknowledge that some of the alkylating esters used in this study are not yet cost-competitive with commodity alcohols. However, their inclusion is not intended as a direct replacement, but rather to demonstrate the broad functional group tolerance and feedstock flexibility of the esterolysis platform. As the reviewer rightly points out, esters may not be universally applicable in all depolymerization contexts. Nevertheless, given their distinct advantages in reaction control, safety,

selectivity, and sustainability, we believe esterolysis offers a versatile and conceptually valuable strategy. It holds strong potential not only for specialized applications but also for scalable industrial implementation in targeted polyester upcycling scenarios.

[1] Shen, Z. Q. et al. CO₂-enhanced PET depolymerization by catalyst free methanolysis. *Process Saf. Environ. Protect.* 188, 230-238 (2024).

[2] Kurokawa, H., Ohshima, M., Sugiyama, K. & Miura, H. Methanolysis of polyethylene terephthalate (PET) in the presence of aluminium triisopropoxide catalyst to form dimethyl terephthalate and ethylene glycol. *Polym. Degrad. Stabil.* 79, 529-533 (2003).

[3] Mishra, S. & Goje, A. S. Kinetic and thermodynamic study of methanolysis of poly(ethylene terephthalate) waste powder. *Polym. Int.* 52, 337-342 (2003).

[4] Du, J. T. et al. ZnO nanodispersion as pseudohomogeneous catalyst for alcoholysis of polyethylene terephthalate. *Chem. Eng. Sci.* 220, 115642 (2020).

[5] Pham, D. D. & Cho, J. Low-energy catalytic methanolysis of poly(ethyleneterephthalate). *Green Chem.* 23, 511-525 (2021).

[6] Laldinpuii, Z. T. et al. Biomass waste-derived recyclable heterogeneous catalyst for aqueous aldol reaction and depolymerization of PET waste. *New J. Chem.* 45, 19542-19552 (2021).

[7] Lalmangaihzuala, S., Laldinpuii, Z. T., Khiangte, V., Lallawmzuali, G. & Vanlaldinpuia, K. Orange peel ash coated Fe₃O₄ nanoparticles as a magnetically retrievable catalyst for glycolysis and methanolysis of PET waste. *Adv. Powder Technol.* 34, 12 (2023).

[8] Tanaka, S., Sato, J. & Nakajima, Y. Capturing ethylene glycol with dimethyl carbonate towards depolymerisation of polyethylene terephthalate at ambient temperature. *Green Chem.* 23, 9412-9416 (2021).

[9] Laldinpuii, Z. T. et al. Methanolysis of PET Waste Using Heterogeneous Catalyst of Bio-waste Origin. *J. Polym. Environ.* 30, 1600-1614 (2022).

[10] Ye, B. Y. et al. Upcycling of waste polyethylene terephthalate to dimethyl terephthalate over solid acids under mild conditions. *Green Chem.* 25, 7243-7252 (2023).

[11] Tang, S. X. et al. Calcined sodium silicate as solid base catalyst for alcoholysis of poly(ethylene terephthalate). *J. Chem. Technol. Biotechnol.* 97, 1305-1314 (2022).

- [12] Tang, S. X. et al. MgO/NaY as modified mesoporous catalyst for methanolysis of polyethylene terephthalate wastes. *J. Environ. Chem. Eng.* 10, 107927 (2022).
- [13] Tang, J. et al. Mechanistic insights of cosolvent efficient enhancement of PET methanol alcohololysis. *Ind. Eng. Chem. Res.* 62, 4917-4927 (2023).
- [14] Liu, M. S., Guo, J., Gu, Y. Q., Gao, J. & Liu, F. S. Versatile Imidazole-anion-derived ionic liquids with unparalleled activity for alcoholysis of polyester wastes under mild and green conditions. *ACS Sustain. Chem. Eng.* 6, 15127-15134 (2018).
- [15] Jiang, Z. Q. et al. Poly(ionic liquid)s as efficient and recyclable catalysts for methanolysis of PET. *Polym. Degrad. Stabil.* 199, 109905 (2022).
- [16] Ma, M. Y. et al. Insights into the depolymerization of polyethylene terephthalate in methanol. *J. Appl. Polym. Sci.* 139, e52814 (2022).
- [17] Li, J. B. et al. Efficient methanolysis of PET catalyzed by nonmetallic deep eutectic solvents. *Ind. Eng. Chem. Res.* 63, 12373-12384 (2024).
- [18] Qu, X. L. et al. Synergistic catalysis of imidazole acetate ionic liquids for the methanolysis of spiral poly(ethylene 2,5-furandicarboxylate) under a mild condition. *Green Chem.* 23, 1871-1882 (2021).

Comments 4: Despite the high catalyst loading and large solvent volume, the depolymerisation of PET remains far too slow considering the 140C reaction time, which raises further concerns about the efficiency of the method.

Response: Thank you for highlighting this important point. We acknowledge that the depolymerization of PET in our system appears relatively slow given the catalyst loading, solvent volume and reaction temperature. To assess efficiency more comprehensively, we compared our method with reported PET methanolysis systems based on three key metrics: reaction temperature, DMT yield, and reaction time (see **Table R6**). While our reaction time is not particularly short, the process remains competitive under comparable temperature conditions. Importantly, our esterolysis system offers clear advantages in both DMT yield and operating temperature, especially compared to

ionic liquid-catalyzed systems. Many of these systems fail to achieve full PET depolymerization, often yielding <91% DMT (**entries 13-18**), and require higher temperatures to reach even moderate conversion levels.

Based on our mechanistic investigations, we identified a simple yet effective strategy to significantly enhance the reaction rate: the introduction of small amounts of methanol or water. We found that adding just 0.6 mmol of methanol to the standard reaction system resulted in complete PET conversion within 40 minutes, delivering DMT and EC in yields of 99% and 85%, respectively (**Figure R8a**). This represents a sixfold reduction in reaction time compared to the original 4-hour process. Similarly, the addition of 0.1 mmol of water increased the DMT yield from 52% to 70% (**Figure R8b**), demonstrating that even minimal water content can significantly accelerate PET depolymerization. This promotional effect extends to ester-based substrates such as DMM, where both methanol and water improved reaction efficiency (**Figures R8c and R8d**). For instance, in the presence of water alone, a 64% yield was achieved within 2 hours, highlighting the robustness of this enhancement. These findings suggest that the incorporation of trace amounts of alcohol and/or water offers a practical means to accelerate esterolysis-based PET depolymerization. This not only improves process efficiency but also provides operational flexibility. In scenarios where alcohol use is restricted due to safety considerations, water can serve as a viable alternative to enhance reaction rates, making this approach particularly appealing for industrial implementation

This work primarily aims to establish esterolysis as a novel and generalizable strategy for polyester conversion, wherein esters act as alternative alkyl sources in place of alcohols during depolymerization. The emphasis of this study is on providing proof-of-concept evidence for this mechanistic pathway, demonstrating its feasibility and potential advantages. We acknowledge that the current system exhibits certain limitations in terms of catalyst efficiency, solvent use, and overall reaction performance, as rightly noted by the reviewers. Future efforts will focus on optimizing the catalytic system and exploring or designing more efficient catalysts for each specific reaction to further shorten the reaction time and enhance overall efficiency. We sincerely thank the reviewers for their insightful comments and constructive suggestions, which have helped us clarify the significance of this approach and improve the manuscript.

Table R6. Methanolysis of PET with reported catalysts.

Entry	Catalysts	Co-solvent	Temp. (°C)	T (h)	Conv. (%)	Yield (%)	Ref.
1	CO ₂	-	220	0.83	100	79.1	1
2	AIP	Toluene	200	2	-	88.5	2
3	Pb(AC) ₂ + Zn(AC) ₂	-	120	2	97.8	97.8	3
4	ZnO nanodispersions	-	170	0.25	97	95	4
5	K ₂ CO ₃	DCM	25	24	100	93.1	5
6	Orange peel ash	-	200	1	-	79	6
7	OPA@Fe ₂ O ₃	-	200	1	100	83	7
8	LiOMe	DMC	65	5	-	91	8
9	Bamboo leaf ash	-	200	2	100	78	9
10	Ti _{0.5} Si _{0.5} O ₂	-	160	2	100	98.2	10
11	Calcined sodium silicate	-	200	0.5	100	95	11
12	MgO/NaY	-	200	0.5	99	91	12
13	ChCl/Zn(OAc) ₂	MeCN	170	1	100	90.1	13
14	[HDBU][Im]	-	140	3	100	75	14
15	PIL-Zn ²⁺	-	170	1	100	90.3	15
16	[HO ₃ S-(CH ₂) ₃ - NEt ₃]Cl[ZnCl ₂] _{0.67}	-	195	0.5	-	78.4	16
17	DBN/Phenol	-	130	1	100	95.3	17
18	[BMIm][OAc]	-	150	4	88.5	41.7	18
19	[EMIm][OAc]	NMP	140	2	100	90	This work
20	[EMIm][OAc]	NMP	140	4	100	99	This work

Figure R8. The effect of MeOH or water content on DMC or DMM-involved PET esterolysis.

Reviewer #3: The author has introduced a novel ester hydrolysis pathway based on an ester exchange mechanism and validated its potential applications. While the innovation is promising, several issues need to be addressed:

Response: We appreciate the reviewer for the positive comments and suggestions for improving this manuscript.

Comments 1: The manuscript utilizes NMP as the solvent but separating it from the product during recovery is challenging. Additionally, the catalyst used in the reaction and the product, ethylene carbonate (EC), are not effectively separated or recovered, making it impractical for waste plastic recycling. The author must address this complication

Response: We thank the reviewer for highlighting the important considerations regarding solvent and catalyst recovery, which are critical for the practical application of plastic recycling strategies.

As noted, NMP is a high-boiling solvent, and its recovery can pose challenges in large-scale processes. However, NMP was used as a typical solvent to evaluate reaction performance and is not essential to the strategy. NMP represents only one of several high-boiling-point solvents typically used in conventional reaction condition optimizations, such as DMF and DMSO, which were also evaluated in our study and found to serve as effective media for esterolysis, affording DMT yields of 89% and 96%, respectively. Notably, a broad range of low-boiling and readily recyclable solvents, including acetonitrile, benzonitrile, 1,4-dioxane, MeCN, toluene, cyclohexane, THF, acetone, and DMC, were also compatible with the esterolysis strategy, delivering DMT yields ranging from 80% to 99% (**Table R7**). More importantly, some carbonates such as diethyl carbonate and diallyl carbonate can also serve directly as solvents to afford the corresponding terephthalate esters. The use of these volatile solvents allows for efficient recovery and reuse, thereby mitigating the environmental impact.

To address scalability and recovery concerns directly, we performed scale-up experiments in which DMC functioned as both the solvent and the methanol source, eliminating the need for external methanol and high-boiling-point solvents such as NMP. These experiments, conducted on a 50 g scale using PET bottle flakes, were designed to further evaluate the efficiency of the PET degradation system and demonstrate its potential for scalability and sustainability. As shown in **Figure R9**, the resulting organic mixture can be readily separated via solid–liquid separation to afford a liquid product mixture

and crude DMT. The crude DMT can be purified by recrystallization to yield high-purity DMT (98%), while the liquid phase can be efficiently recycled through evaporation to recover DMC. We further analyzed the recovered DMC using ^1H and ^{13}C NMR spectroscopy, confirming its effective recovery and showing only trace residual methanol, thus supporting the recyclability of the solvent in this system.

Furthermore, our poly(ionic liquid) catalyst is compatible with recyclable solvents such as DMC and MeCN (**Table R8**), which facilitates both catalyst reuse and product isolation, enhancing the practical viability of the system.

In summary, our method is not limited to NMP and can be effectively implemented with low-boiling, recoverable solvents and recyclable catalysts, addressing the reviewer's concern regarding solvent and product separation in scalable plastic recycling.

Table R7. Comparison of reaction solvent.

Entry	Cat.	Load.	Temp.	Sol.	DMT Yield (%) ^b
1	[Emim][OAc]	20 mol%	140 °C	DMSO	89
2	[Emim][OAc]	20 mol%	140 °C	NMP	99
4	[Emim][OAc]	20 mol%	140 °C	DMF	96
5	[Emim][OAc]	20 mol%	140 °C	Benzotrifluoride	94
6	[Emim][OAc]	20 mol%	140 °C	1,4-Dioxane	92
7	[Emim][OAc]	20 mol%	140 °C	MeCN	91
8	[Emim][OAc]	20 mol%	140 °C	Toluene	91
10	[Emim][OAc]	20 mol%	140 °C	Cyclohexane	85
11	[Emim][OAc]	20 mol%	140 °C	THF	80
12	[Emim][OAc]	20 mol%	140 °C	Acetone	92
13	[Emim][OAc]	20 mol%	140 °C	Chlorobenzene	9
14	[Emim][OAc]	20 mol%	140 °C	Bromobenzene	-
15	[Emim][OAc]	20 mol%	140 °C	Chloroform	-
16	[Emim][OAc]	20 mol%	140 °C	DMC	99

Catalytic system and reaction conditions exploration. ^a Standard reaction conditions: PET (1 mmol), DMC (4 mmol), [Emim][OAc] (20 mol%), and solvent (3 mL) at 140 °C for 4 h. ^b Yields were mesitylene by GC analysis with mesitylene as internal standard.

Figure R9. Solvent recovery experiments: (a) Separation of solvent and recovery of products; (b) ^1H NMR spectrum of the recycled DMC; (c) ^{13}C NMR spectrum of the recycled DMC.

Table R8. PILs-catalyzed PET esterolysis condition with DMC^a

Entry	Cat.	Load.	Temp.	Sol.	DMT (%) ^b	Yield (%)	EC (%) ^b	Yield (%)
1	PIL-OAc	0.05g	150 °C	NMP	95	70		
2	PIL-OAc	0.05g	160 °C	NMP	99	78		
3	PIL-OAc	0.05g	170 °C	NMP	99	45		
4	PIL-OAc	0.1g	160 °C	NMP	99	60		
5	PIL-OAc	0.03g	160 °C	NMP	97	71		
6	PIL-OAc	0.05g	160 °C	MeCN	99	81		
7	PIL-OAc	0.05g	160 °C	Acetone	99	39		
8	PIL-OAc	0.05g	160 °C	DMC	99	52		

Catalytic system and reaction conditions exploration. ^a Standard reaction conditions: PET (0.192g, 1 mmol structural unit), DMC (4 mmol), and solvent (3 mL) at 140 °C for 4 h. ^b Yields were determined by ^1H NMR analysis with mesitylene as internal standard.

Revised Manuscript and Supporting Information Details:

Revised manuscript, Bottom of Page 2: Notably, the strategy exhibits broad solvent compatibility, affording DMT in high yields (89-96%) across a range of common solvents instead of NMP, including DMSO, DMF, MeCN, 1,4-dioxane, acetone, and toluene.

Top of Page 3: Notably, such carbonates, including DMC, diethyl carbonate, and diallyl carbonate, can readily function as both solvents and alkylating sources, enabling high yields of the corresponding terephthalate esters while simultaneously avoiding the use of toxic external alcohols and solvent like NMP.

Bottom of Page 4: To demonstrate the potential for industrial scalability, gram-scale reactions were conducted using 30 g and 50 g of raw PET bottle flakes in either NMP or DMC as the reaction medium. After 4 hours at 140 °C, the reactions yielded DMT in 97% and 98%, and EC in 81% and 85%, respectively.

Revised supporting information, Page 16, Figure S5: To validate the effectiveness of this strategy, we conducted a scale-up experiment on a 50 g scale using PET bottle flakes, with DMC serving as both the solvent and methanol source, without the need for additional methanol. As shown in Figure S5a, the post-reaction mixture could be readily separated via solid–liquid separation, yielding a liquid product phase and crude DMT. The crude DMT was purified by recrystallization to afford high-purity DMT (98%), while the liquid phase was efficiently recycled by rotary evaporation to recover DMC, leaving only trace amounts of methanol (Figure S5b and 5c)..

Comments 2: Can the catalyst be recycled in this reaction? This is crucial for waste plastic recycling.

Response: We thank the reviewer for highlighting this important aspect. The recyclability of the catalyst is indeed crucial for the sustainability and practical implementation of plastic depolymerization strategies. As noted, the recovery of homogeneous catalysts such as [EMIm][OAc] presents well-known challenges, including separation difficulties and potential contamination of products, particularly in the context of ionic-liquid-promoted polyester recycling.

To address this, we developed a recyclable alternative: a poly(ionic liquid) (PIL)-based catalyst, PIL-OAc, synthesized from 1-vinyl-3-methylimidazolium acetate, zinc acetate, and 1,4-divinylbenzene (Polym. Degrad. Stabil. 2022, 199, 109905). This macromolecular catalyst combines the tunable functionality of ionic liquids with the mechanical stability and easy separability of polymers. Initial screening confirmed that PIL-OAc exhibited excellent catalytic activity, affording DMT and EC in yields of 99% and 78%, respectively (Table R9).

Table R9. PILs-catalyzed PET esterolysis condition with DMC^a

Entry	Cat.	Load.	Temp.	Sol.	DMT (%) ^b	Yield EC (%) ^b	Yield
1	PIL-OAc	0.05g	150 °C	NMP	95	70	
2	PIL-OAc	0.05g	160 °C	NMP	99	78	
3	PIL-OAc	0.05g	170 °C	NMP	99	45	
4	PIL-OAc	0.1g	160 °C	NMP	99	60	
5	PIL-OAc	0.03g	160 °C	NMP	97	71	
6	PIL-OAc	0.05g	160 °C	MeCN	99	81	
7	PIL-OAc	0.05g	160 °C	Acetone	99	39	
8	PIL-OAc	0.05g	160 °C	DMC	99	52	

Catalytic system and reaction conditions exploration. ^a Standard reaction conditions: PET (0.192g, 1 mmol structural unit), DMC (4 mmol), and solvent (3 mL) at 140 °C for 4 h. ^b Yields were determined by ¹H NMR analysis with mesitylene as internal standard.

Recycling experiments were conducted to evaluate the reusability of PIL-OAc. The catalyst was readily recovered from the reaction mixture through phase separation, followed by water washing and drying. Following recycling, the DMT yield remained nearly unchanged, while a moderate decrease

in EC yield was observed (**Table R10**). Although further refinement of the catalyst synthesis and enhancement of its long-term stability are needed, the demonstrated reusability of PIL-OAc highlights its potential for scalable and sustainable application in esterolysis of waste polyesters.

Moreover, while [EMIm][OAc] functions as a general and effective catalyst across the diverse esterolysis routes explored in this study, further tailoring of catalyst systems to suit specific reaction conditions and target products will be important for maximizing efficiency and sustainability. Although optimization of catalyst formulation and recovery protocols is still underway, our current results demonstrate the feasibility of catalyst reuse, laying a solid foundation for the development of scalable and recyclable catalytic systems for waste polyester upcycling.

Table R10. Recyclability Test of PIL-OAc

Cycle Number	Load.	Temp.	Sol.	DMT Yield (%) ^b	EC Yield (%) ^b
1	0.05g	160 °C	NMP	99	72
2	0.05g	160 °C	NMP	97	63

Catalytic system and reaction conditions exploration. ^a Standard reaction conditions: PET (0.192g, 1 mmol structural unit), DMC (4 mmol), and solvent (3 mL) at 140 °C for 4 h. ^b Yields were determined by ¹H NMR analysis with mesitylene as internal standard.

Here are the details:

Revised manuscript, Bottom of Page 4: To address the issue of catalyst recyclability, we developed a polymeric ionic liquid catalyst (PIL-OAc), whose catalytic performance and reuse capability are summarized in Tables S6 and S7. These results further enhance the practical potential of this esterolysis strategy for sustainable plastic waste valorization.

Revised supporting information, Page 18, Table S6: In response to the challenges associated with recycling [EMIm][OAc], we explored the use of poly(ionic liquid)s and successfully designed and synthesized a novel catalyst, PIL-OAc, using p-divinylbenzene, zinc acetate, and 1-vinyl-3-methylimidazolium acetate as monomers, with AIBN as the initiator and MeCN as the solvent. To optimize the reaction conditions, a preliminary screening of catalyst loading, temperature, and solvent was performed. The results demonstrated that the synthesized PIL-OAc exhibited excellent catalytic

activity, enabling complete depolymerization of PET. Notably, when NMP was used as the solvent at 160 °C, DMT and EC were obtained in yields of 99% and 78%, respectively. Notably, efficient PET conversion can be achieved using low-boiling-point solvents such as acetonitrile, yielding DMT and EC in 99% and 81%, respectively.

Page 19, Table S7: The PIL-OAc catalyst can be readily separated from the reaction system after the reaction. Even after three recycling cycles, it still enables efficient conversion of PET to DMT with a high yield of 97%. Although a slight decrease in DMT yield was observed upon extended recycling, the successful implementation of this process provides strong evidence supporting the recyclability of this catalytic system.

Comments 3: Why was [Bmim]Br chosen over [Bmim]OAc in the experiments detailed in Figure 3b? Figure 3 explores a broad range of reactants, yet the yields of the corresponding ester exchange products are not indicated. Please include this information.

Response: We thank the reviewer for this thoughtful question. In the scope expansion experiments shown in **Figure 3b**, we observed that certain methyl esters exhibited lower reactivity compared to carbonates and dimethyl malonate. To address this, we employed [Bmim]Br instead of [Bmim]OAc. The choice was based on the higher nucleophilicity of bromide anions, which facilitates more efficient cleavage of relatively inert carboxylic esters and promotes alkyl group transfer to PET.

We appreciate the reviewer's suggestion to include the yields of the corresponding ester exchange products. While we agree that such quantitative data would be informative, the reactions in **Figure 3b** involve complex mixtures of multiple ester substrates. Accurate separation and quantification of each product using standard techniques (e.g., GC, HPLC, NMR) are complicated by overlapping peaks and similar physicochemical properties. To overcome this, we conducted LC-MS analysis on 12 representative carboxylic esters containing C, Ti, Si, and P elements (see **Supporting Information, Section 11**), including dimethyl malonate, methyl trifluoroacetate, titanium methoxide, tetramethoxysilane, and trimethyl phosphate. The EG-derived products predominantly fall into two categories: (1) monoesterification of EG with monoesters, and (2) diesterification of EG with dicarboxylic esters.

While we fully acknowledge the importance of reporting individual product yields, our primary goal in this study is to establish a generalizable esterolysis strategy. We believe that the combination of LC-MS data, representative substrate conversion analysis, and demonstrated product diversity sufficiently supports the efficacy and broad applicability of the method. We respectfully note that exhaustive deconvolution of product yields from multicomponent mixtures, while important, falls beyond the intended scope of this manuscript. Future work will focus on in-depth mechanistic studies and yield optimization for specific esterolysis reactions, as well as improving catalyst stability and recyclability to further enhance the sustainability and scalability of the strategy.

Comments 4: As mentioned by the author, the OAc anion initiates the activation of the ester hydrolysis process by activating methanol produced from DMC decomposition. However, this resembles typical alcoholysis mechanisms. The author should further elaborate on the distinctions between these pathways to illustrate that the proposed ester hydrolysis pathway is genuinely innovative.

Response: We sincerely thank the reviewer for this thoughtful and constructive comment. To clarify the distinction between our proposed esterolysis mechanism and conventional alcoholysis pathways, we have conducted an in-depth mechanistic analysis supported by experimental data and theoretical calculations.

In addition to the classical methanolysis route, our findings reveal an alternative pathway in which DMT is generated via direct methylation of carboxylic acid intermediates formed through hydrolysis. This mechanism is fundamentally different from conventional methanolysis, where methanol directly cleaves the ester bonds in polyethylene terephthalate to yield DMT and EG typically under strictly anhydrous and relatively harsh conditions.

While classical methanolysis typically involves the direct attack of methanol on ester bonds under strictly anhydrous and relatively harsh conditions, our results suggest a mechanistically distinct, water-tolerant pathway. In this system, DMC serves not merely as a methanol source but as a methylating agent that promotes an alternative hydrolysis–methylation route for DMT formation. Specifically, carboxylic acid intermediates generated via hydrolysis subsequently undergo methylation with DMC, bypassing the need for direct alcoholysis by methanol.

Figure R10. The effect of MeOH or water content on DMC or DMM-involved PET esterolysis.

The following lines of evidence support our proposed mechanism:

- (1) Promotional effect of water:** In contrast to traditional views where water is regarded as detrimental due to its competition with methanol for ester linkages, which often leads to incomplete depolymerization or the formation of undesired hydrolysis byproducts, our results demonstrate that water can actively promote the esterolysis reaction involving typical carboxylic esters such as dimethyl malonate. Unlike systems where carbonates serve as the alkyl source, the addition of water significantly enhances both the depolymerization rate and the DMT yield in DMM-involved esterolysis processes (Figure R10). Under identical reaction times, the presence of water enhances the DMT yield significantly, increasing it from 25% to 64%. This finding underscores the constructive role of water in facilitating the sequence of hydrolysis followed by methylation.
- (2) Direct methylation of carboxylic acid intermediates:** Experimental results show that certain PET-derived compounds such as terephthalic acid (TPA), monomethyl terephthalate (MMT), hydroxymethylphthalic acid (HMPTA), bis(2-hydroxyethyl) terephthalate (BHET), and various

substituted aromatic carboxylic acids, including benzoic acids with different functional groups, fused ring systems, and heteroaromatic derivatives, undergo direct methylation with DMC (**Figure R11 and S16**). In contrast, no methyl transfer occurs when methanol is used as the methylating agent under comparable conditions (Figure R11B, reaction 2). Furthermore, the methylation reaction with DMC proceeds significantly faster than other competing pathways, indicating that this transformation is highly efficient relative to other methylation reactions. These findings suggest that if carboxylic acid intermediates are generated through hydrolysis in the system, they preferentially undergo methylation with DMC rather than reacting with methanol, highlighting a mechanistic divergence from conventional methanolysis.

Figure R11. Direct methylation reactions of PET depolymerization-derived model compounds.

(3) **Theoretical validation via DFT calculations:** To further verify the feasibility of direct methylation between the carboxyl group of intermediates and DMC, we performed Density Functional Theory (DFT) calculations using benzoic acid and DMC as model reactants (**Figure R13**). The DFT results revealed that both investigated pathways proceed through distinct transition states to accomplish the transesterification reaction. The energy barrier of the transition state in path 1 was slightly higher than that of path 2, with calculated values of 32.7 and 27.8 kJ/mol, respectively. Moreover, the energy level of the intermediate IM2 following the transition state in path 2 was significantly lower than that of IM1 in path 1. These results indicate that path 2, in which the carboxyl group of the intermediate attacks the methyl group of DMC, is not only feasible but also the energetically preferred route for transesterification.

Figure R13. DFT-supported pathway for direct methylation of carboxylic acids.

In summary, the esterolysis process proceeds through a dual-pathway mechanism. One pathway resembles classical methanolysis, while the other involves a water-mediated sequence in which ester bonds are hydrolyzed to form carboxylic acid intermediates, which are subsequently methylated to produce DMT. This alternative hydrolysis–methylation route is proposed here for the first time and is supported by experimental evidence and DFT calculations. It represents a mechanistically distinct and water-tolerant approach to DMT formation, challenging the conventional understanding of polyester depolymerization.

Here are the details:

Revised manuscript, Bottom of Page 5: The results revealed that even small additions of MeOH markedly accelerated the esterolysis process, irrespective of whether DMC or DMM was used as the transesterification reagent. Similarly, low concentrations of H₂O enhanced the reaction efficiency in DMC-mediated systems and exhibited an even more pronounced effect in DMM-involved esterolysis. These findings indicate that both H₂O and in situ generated MeOH are pivotal in initiating the depolymerization process.

Middle of Page 6: Notably, carboxylic acids (TPA and MMT), derived from PET hydrolysis, reacted with DMC to afford DMT in yields of 49% and 99%, respectively, indicating that PET hydrolysis to carboxylic acid intermediates followed by consecutive methylation provides a possible and viable transformation pathway.

Bottom of Page 6: While methanolysis appears to be the predominant pathway, supporting evidence also points to a concurrent esterolysis route involving carboxylic acid esters, as illustrated in Figure S18.

Revised supporting information, Page 34, Figure S18: Based on experimental evidence for the promoting effect of water on PET esterolysis involving carboxylic acid ester substrates and the methylation of carboxylic acid intermediates (as illustrated in Figures 5B, 5C, S9, and S16), we propose that although methanolysis remains the dominant pathway in the system, DMM-involved PET esterolysis may also proceed via hydrolysis and involve direct methylation of the carboxylic acid intermediates by DMM.

Figure S18. The possible reaction mechanism of DMM-involved PET esterolysis catalyzed by [EMIm][OAc].

Comments 5: Minor errors: capitalization issues in reference titles. Please review and rectify these errors.

Response: Thank you for raising this important point. In response to the reviewer's recommendations, we have carefully reviewed and revised the formatting of all references in both the manuscript and the Supporting Information. All capitalization inconsistencies have been addressed and corrected to ensure full compliance with the journal's style guidelines.

Reviewer #1: Having revised the updated version of the manuscript, all the points I initially indicated have been satisfactorily addressed. I therefore consider that the manuscript could be accepted.

Response: We sincerely thank you for your time and effort in reviewing our manuscript. Your insightful comments and suggestions have played an important role in improving the quality and clarity of our work. We are truly honored by your positive recommendation and greatly appreciate your recognition and support.

Reviewer #2: Considerable effort has been made to address concerns, however while the effort is commendable, the main point here seems to be lost. These are not interesting molecules (the products) which can be easily synthesized in the lab. The fact that esterolysis is 'new' does not make it interesting to the broad readership of Nature Communications unless it looks like this might be a practical methodology at scale. Then it is very interesting. This does not mean that all the scale-up issues need to be worked out now - however there are so many barriers to adoption in any practical sense here (given how cheap PET and DMT are) that the excess reagents, high loadings of catalyst etc make the process difficult to see as being useful. For example, the statement in the rebuttal 'in contrast, esterolysis in our study operates efficiently with 3-8 equivalents of ester, thus requiring a lower overall reagent input' is fine in the lab, but are we going to use 3-8 equivalents of ester (made themselves from alcohols and acids) on multi-tonne scale? Using DMC is nice, but also well known in biodiesel production, and its more expensive than methanol. The reactions were also less efficient in it. The solvent can be recycled, but you can clearly see MeOH in the recycled product by ¹H NMR (not exactly a sensitive technique). The catalyst loading problem is not solved in my view. The authors have provided a technicality which actually takes the methodology further from being useful. They make a polymer bound catalyst, requiring investment, reagents etc. This is fine. But ca. 25 wt% of this is required to catalyze the reaction, at higher temperatures than before. They recycle it 3 times, and EC yields are already dropping on the first recycle and both DMT and EC are lower on the third. This is clearly not a strategy that would be adopted at scale. What industry would invest in making a catalyst (more complex than simple systems available for DMT synthesis) which is already less efficient after one recycle? This is a technicality - it allows the authors to claim that a recyclable catalyst is available, but which in fact does not make the system any more practical. Heating at high temperatures for long times is still a problem too. While the authors have made attempts to remedy the issues, I feel the process as it stands is just not practical enough (by a wide margin) to be of general interest.

Response: We sincerely thank the reviewer for their thoughtful and constructive feedback. We appreciate the recognition of our efforts to address earlier concerns and the emphasis on key limitations related to catalyst loading, reagent usage, reaction conditions, and the broader issue of practical implementation. These are indeed critical factors requiring further development, and we fully agree that the current system does not yet meet the criteria for industrial applicability. We have carefully

considered each of these points in our revision. Below, we provide detailed responses to the key concerns you raised:

On industrial relevance and scalability:

We fully acknowledge that this study represents a proof-of-concept and does not claim industrial readiness. Our goal is to demonstrate a conceptual advancement in PET depolymerization via a metal-free esterolysis strategy under relatively mild conditions (100–140 °C), in contrast to many conventional glycolysis and methanolysis protocols. While the polymer-supported catalyst remains at an early stage, it was designed to explore the feasibility of heterogeneous catalysis with potential relevance to future continuous-flow systems. In response to the reviewer's concern, we have revised the manuscript to remove language suggesting immediate industrial scalability and have reframed the discussion to better reflect the fundamental scope of the work. We agree that addressing scale-up, cost, and process efficiency is essential for practical implementation, and these will be important directions for future research.

On the use of ester reagents (3–8 equivalents):

We agree that reagent excess poses a significant challenge for scaling any chemical process. In this study, the quantities were chosen to explore substrate generality and mechanistic behavior, rather than to optimize process efficiency. Many of the esters used (e.g., methyl acetate) can be sourced from low-cost, renewable, or even waste-derived feedstocks, which may help mitigate concerns related to cost and availability. For example, DMC can be synthesized from CO₂ and methanol (CO/CO₂ + H₂), providing a pathway to incorporate renewable carbon. Moreover, the total reagent input remains lower than in many reported PET solvolysis systems that rely on alcohol-based nucleophiles. Nevertheless, we recognize that reagent economy remains a limitation. We have accordingly revised the manuscript to better reflect the proof-of-concept nature of this work and to avoid overstatements related to industrial implementation.

On catalyst recyclability and practical utility:

We appreciate the reviewer's assessment regarding catalyst performance. As correctly noted, the recyclability of the polymeric catalyst remains limited, with reduced yields upon repeated use. This

limitation is clearly acknowledged in the revised manuscript. The catalyst design in this study represents a preliminary step toward heterogeneous systems for esterolysis, and further improvements in stability, turnover, and compatibility are indeed necessary. Our intention is not to present this as a finalized or scalable solution, but rather to demonstrate that esterolysis of PET can be realized under metal-free, heterogeneous conditions. We hope this serves as a stepping stone toward more robust systems in the future.

In summary, in response to the reviewer's suggestion, we have revised the manuscript to more accurately reflect the proof-of-concept nature of this study and have removed or reframe industrialization-related claims throughout. We believe these changes provide a clearer and more accurate representation of the contribution. While we fully recognize that the current system has practical limitations, we hope that the mechanistic insights and conceptual framework can provide a useful foundation for further development toward scalable and sustainable PET upcycling. Once again, we sincerely thank the reviewer for their thoughtful critique, which has greatly helped improve the clarity, focus, and scientific framing of our manuscript.

Here are the details:

Revised manuscript, Bottom of Page 2:

Original: To demonstrate the potential for industrial scalability, gram-scale reactions were conducted using 30 g and 50 g of raw PET bottle flakes in either NMP or DMC as the reaction medium.

Revised: (removing industrialization-related information): Gram-scale reactions were further conducted using 30 g and 50 g of raw PET bottle flakes in either NMP or DMC as the reaction medium.

Original: These results further enhance the practical potential of this esterolysis strategy for sustainable plastic waste valorization.

Revised: (removing industrialization-related information): These results demonstrate the effectiveness of this esterolysis strategy in achieving plastic waste valorization.

Revised manuscript, Discussion:

Original: In this study, we presented an efficient esterolysis strategy for upcycling waste polyester into valuable esters, addressing critical challenges in industrial polyester recycling, such as incomplete conversion and harsh reaction conditions.

Revised: In this study, we presented an efficient esterolysis strategy for upcycling waste polyester into valuable esters via transesterification, offering an alternative approach for polyester valorization.

Original: Future efforts should focus on developing highly efficient and easily recyclable heterogeneous catalysts to facilitate industrial implementation.

Revised: Future research should prioritize the design of highly efficient, task-specific, and easily recyclable heterogeneous catalysts to expand the applicability and enhance the sustainability of this approach.

Reviewer #3: We have carefully reviewed the revised manuscript and the authors' responses to our comments. Overall, the authors have addressed the majority of the concerns raised in the previous review, and the revisions have improved the clarity and rigor of the work. Regarding Question 3 (on quantification in substrate scope evaluation), while the authors were unable to resolve the issue of product quantification definitively, they have provided reasonable justification for their approach and clarified their perspective. Although further experimental validation would have strengthened the study, their explanation is acceptable in the current context. Given the authors' efforts in addressing the reviewers' feedback and the manuscript's scientific merit, we recommend acceptance of the paper in its present form.

Response: Thank you very much for your careful review and valuable comments on our manuscript, which have significantly contributed to improving the clarity and overall quality of our work, especially regarding key professional aspects such as catalyst recycling, solvent selection, product scope evaluation, and mechanistic validation. We are deeply honored by your positive evaluation and sincerely appreciate your insightful and constructive advice.